# Scalable Medical Multimodal Fusion via Symmetric Consistency Modeling

**Xiaowen Sun** [1 2]  **Hui Liu** [1 2]  **Gongguan Chen** [1 2]  **Ning Mao** [3 4 5]

## Abstract

Medical diagnosis tasks often rely on heterogeneous information from multiple sources, such as medical images and clinical text. Multimodal fusion is therefore essential for improving classification performance and robustness. However, most existing methods assume a fixed and known modality set, making them less effective when the number or composition of modalities changes. To address this limitation, we propose a modality-agnostic medical multimodal fusion framework that can naturally accommodate an arbitrary number of input modalities. At the coarse-grained modality level, we represent each modality's estimation of latent semantics as an uncertainty-aware probability distribution, and impose symmetric consistency constraints to achieve global cross-modal semantic alignment. At the fine-grained token level, we further design a token-level consistency constraint based on linear reconstruction. This constraint enables structured mutual verification of local semantics across modalities. Finally, for multimodal fusion, we adopt a multi-view consistency strategy to obtain a unified representation for diagnosis prediction. In particular, each modality is sequentially treated as a conditional view to suppress noise in the remaining modalities and extract shared semantics. Extensive experiments on five public and self-constructed multimodal medical datasets demonstrate the effectiveness and scalability of the proposed approach. Code is available at https://github.com/xxiaowSun/SMMF.

## 1. Introduction

With the rapid development of medical imaging technologies and clinical information systems, data available in medical diagnosis show clear multimodal characteristics. For example, the same patient often has multiple medical images, clinical text descriptions, and other auxiliary examination data. These modalities describe the patient's health status from different perspectives and contain complementary medical semantic information(Daniaux et al., 2023; Gong et al., 2025). How to effectively fuse multimodal medical data is crucial for improving disease classification accuracy.

Existing studies mainly focus on multimodal feature fusion and representation learning(Li & Tang, 2024; Shaik et al., 2024). Related methods can be roughly divided into feature-level fusion, cross-modal representation alignment, and attention modeling. Early works often adopt feature-level fusion strategies, where features from different modalities are concatenated or combined by weighted summation(Jiao et al., 2024). Later, some studies introduce cross-modal representation alignment mechanisms by mapping different modalities into a shared semantic space(Qian et al., 2025). Consistency constraints are imposed in this space to reduce the impact of distribution differences across modalities on classification performance. In recent years, with the development of deep learning, attention mechanisms have been widely used in multimodal medical modeling(Song et al., 2022; Sun et al., 2024). They learn importance weights across or within modalities to adaptively focus on key information.

Although existing methods have achieved certain progress in multimodal medical classification tasks, they still face several challenges in more complex semantic modeling and practical scenarios. First, although prior works introduce fine-grained features for cross-modal interaction, they often treat them as discriminative information for direct fusion, lacking explicit constraints on cross-modal local semantic structure consistency(Tao et al., 2024; Qian et al., 2025). Second, different medical modalities show significant inconsistency in expressing the same semantics. Most existing

[1]School of Computer and Artificial Intelligence, Shandong University of Finance and Economics, Jinan, China [2]Shandong Key Laboratory of Lightweight Intelligent Computing and Visualization for Digital Economy, Shandong University of Finance and Economics, Jinan, China [3]Big Data and Artificial Intelligence Laboratory, Qingdao University, Qingdao, China [4]Department of Radiology, Yantai Yuhuangding Hospital, Yantai, China [5]Shandong Provincial Key Medical and Health Laboratory of Intelligent Diagnosis and Treatment for Women's Diseases, Yantai Yuhuangding Hospital, Yantai, China. Correspondence to: Hui Liu <liuh_lh@sdufe.edu.cn>.

*Proceedings of the 43rd International Conference on Machine Learning*, Seoul, South Korea. PMLR 306, 2026. Copyright 2026 by the author(s).

methods treat each modality as a deterministic vector(Lan et al., 2024). This fails to effectively model modality-level semantic uncertainty and limits robustness on complex clinical data. Finally, existing methods usually assume that the number and structure of modalities are known and fixed during model design(Yu et al., 2025). Their fusion strategies and loss formulations are difficult to naturally extend to an arbitrary number of modalities, which significantly constrains scalability and generalization ability.

To address the above issues, this paper constructs a unified multimodal fusion framework from three aspects: representation modeling, consistency constraints, and fusion strategy. First, to handle the significant differences in reliability when different modalities express the same medical semantics, uncertainty modeling is introduced at the semantic encoding stage. Each modality's estimation of latent medical semantics is represented as a probability distribution with uncertainty, enabling adaptive adjustment of alignment strength during modality-level alignment. Second, to address the difficulty of existing methods in capturing fine-grained cross-modal semantic consistency, this paper further introduces a bidirectional consistency constraint based on deterministic linear reconstruction at the token level. This constraint characterizes mutual verification relationships of local semantic representations across different modalities at the structural level. It avoids instability caused by relying solely on attention weights or similarity metrics. Finally, during multimodal fusion, this paper adopts a multi-view consistency perspective, where each modality is sequentially treated as a conditional view to remove noise from the representations of the remaining modalities. This process systematically mines shared and common semantic information across different modalities. The proposed fusion strategy relies only on the aligned modality representations and imposes no restriction on the number of modalities, allowing natural extension to an arbitrary number of modality inputs. The main contributions of this paper are summarized as follows:

- We propose a modality-level consistency alignment method based on semantic uncertainty modeling. It represents different modalities' expressions of latent medical semantics as probability distributions and achieves global cross-modal semantic alignment through statistical consistency constraints.

- We propose a token-level bidirectional consistency constraint based on deterministic linear reconstruction. It characterizes semantic mutual verification relationships from a fine-grained structural perspective.

- We construct a multi-view constrained multimodal fusion method for medical data. This method enables robust and scalable medical diagnosis classification.

## 2. Related Work

### 2.1. Medical Multimodal Fusion

The core goal of medical multimodal classification is to jointly model medical information from different sources to improve diagnostic prediction accuracy and robustness(Bannur et al., 2023). Early studies often adopt feature-level fusion paradigms, where features from different modalities are concatenated or combined by weighted summation before being fed into a classifier(Kumar et al., 2024). Parisot et al.(Parisot et al., 2018) integrate imaging features and phenotypic information into a population graph framework and use graph convolution to propagate information across subjects, laying the foundation for graph-based learning of medical multimodal data. Subsequently, many works follow the population graph paradigm and further improve multimodal brain disease prediction on datasets such as ABIDE(Di Martino et al., 2014) and ADNI(Sandeep et al., 2017). For example, fusion effectiveness is enhanced through stronger graph structure modeling or end-to-end multimodal graph learning(Ali et al., 2025; Song et al., 2024). Beyond population graph fusion, medical vision-language joint modeling has become an important research direction in recent years. Such methods exploit the natural pairing between images and text and perform fusion through contrastive learning or cross-attention mechanisms(Chen et al., 2023). For example, ConVIRT(Zhang et al., 2022) learns transferable representations from paired image-text data through contrastive learning. GLoRIA(Huang et al., 2021) emphasizes fusion at both global and local levels. MGCA(Wang et al., 2022) further achieves image-text fusion at multiple granularities, thereby improving generalization in downstream medical classification. At the same time, recent studies have begun to emphasize uncertainty in multimodal scenarios. For example, DyCON(Assefa et al., 2025) introduces uncertainty-aware mechanisms into consistency learning and contrastive learning. BEAR(Yang et al., 2025) also treats uncertainty as a core issue in multimodal understanding. These studies promote the evolution of medical multimodal fusion from simple combination to interactive joint learning. However, most of these fusion methods rely on predefined modality combinations and struggle to maintain fusion quality while achieving good modality scalability when the number of modalities varies in real clinical settings.

### 2.2. Cross-modal Alignment and Consistency Modeling

Cross-modal alignment aims to alleviate distribution discrepancies and semantic heterogeneity across modalities, enabling different modalities to obtain fusion-ready semantic representations in a shared space(Huang et al., 2025; Kapadnis et al., 2024). A representative line of work imposes consistency constraints within a shared semantic space(Phan et al., 2024). MMGL(Zheng et al., 2022) proposes end-to-

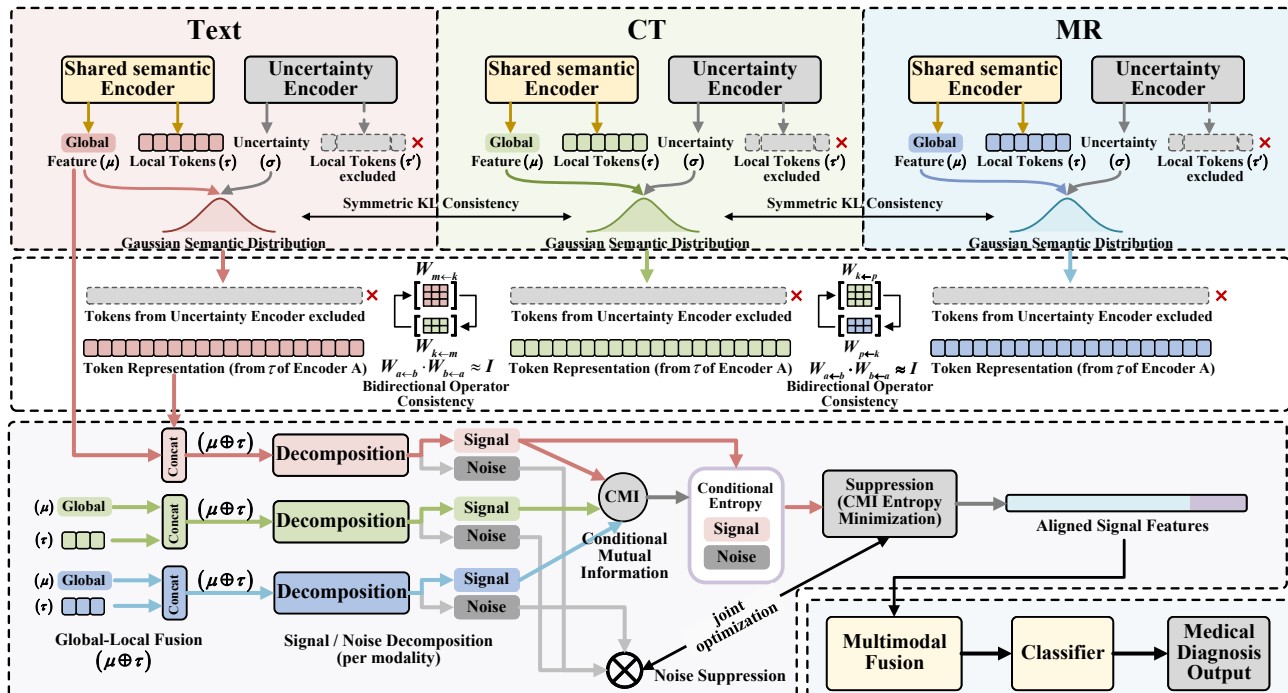

*Figure 1.* Overview of the proposed modality-agnostic multimodal framework for medical diagnosis. The goal of the framework is to achieve robust and scalable multimodal fusion by progressively establishing cross-modal consistency. Top: we perform hierarchical alignment by enforcing symmetric distribution consistency at the modality level. Middle: we further impose bidirectional linear reconstruction consistency at the token level. Bottom: the aligned representations are then fused via a multi-view strategy to produce a unified multimodal feature for final classification.

end adaptive graph learning with modality-aware aggregation to model modality correlations and performs disease prediction without manually fixing graph structures. EV-GCN(Huang & Chung, 2020) learns uncertain population graph structures through variational edge modeling to improve generalization and robustness on multimodal medical data. The neuroimaging field has also gradually focused on stronger structural inductive biases. For example, BrainGNN(Li et al., 2021) constructs brain graphs from fMRI data and introduces ROI-aware designs in graph convolution and pooling layers. In addition, cross-modal attention mechanisms have been widely adopted in recent years. EFNet(Sun et al., 2022) designs an event-image cross-modal attention module that allows the event branch to attend to relevant features while filtering noise. Wu et al.(Wu et al., 2023) analyze the role of shared priors in Bayesian data fusion and study fusion performance under different numbers of cooperative agents and prior types. Yi et al.(Yi et al., 2019) further derive detailed heterogeneous fusion equations for the probability hypothesis density (PHD) filter and the multi-Bernoulli (MB) filter, and propose a centralized network structure. However, most existing alignment methods rely on similarity metrics or attention weights for token-level interaction(Li et al., 2023; Zhou et al., 2025). They lack structured mutual verification of bidirectional explanatory relationships, such as viewing modality A from B

or modality B from A.

## 3. Method

This paper focuses on medical diagnosis tasks with multimodal medical data and constructs an end-to-end framework for multimodal representation learning and fusion. Given patients' multimodal medical data, the model first extracts features from each modality using encoders. It jointly models global semantic representations, local token representations, and corresponding uncertainty information within a unified semantic space. On this basis, consistency across modalities is constrained at both the modality level and the token level. This ensures reliable and stable cross-modal semantic alignment. The model then fuses semantic information from all modalities and feeds the fused multimodal representation into a classifier to predict medical diagnosis outcomes.

### 3.1. Modality-level Consistency Alignment

In multimodal tasks, different modalities can be viewed as distinct observation perspectives of the same latent semantic state. Modalities often vary in reliability when describing the same semantics. To address this issue, this paper introduces a modality-level bidirectional consistency alignment

mechanism based on uncertainty mutual verification.

Let the semantic representation of sample $i$ be denoted as $s_i$, and let the $m$-th modality corresponding to this sample be denoted as $z_i^{(m)} \in \mathbb{R}^d$. This paper adopts a dual-encoder architecture for each modality to model $z_i^{(m)}$. As shown in Figure 1, both encoders extract global features and local features from the modality input to characterize semantic information at different levels. Specifically, the global features output by the first encoder are used as shared semantic features $\mu_i^{(m)}$, while the global features output by the second encoder are used as uncertainty vectors $\sigma_i^{(m)}$. Based on these global outputs, the estimation of the latent semantic variable $s_i$ by the $m$-th modality is modeled as:

$$q_i^{(m)}(s_i) = \mathrm{N}(s_i; \mu_i^{(m)}, diag((\sigma_i^{(m)})^2)). \tag{1}$$

Under this modeling, semantic characterizations of the same sample from different modalities are viewed as statistical measurements of the same latent variable. If two modalities are semantically consistent, the semantic distributions inferred from modality $m$ and modality $k$ should be close in a statistical sense. Based on this idea, this paper adopts symmetric KL divergence as a modality-level bidirectional consistency constraint:

$$L_{bi-align}^{m,k} = E[KL(q_i^{(m)}||q_i^{(k)}) + KL(q_i^{(k)}||q_i^{(m)})]. \tag{2}$$

Since a diagonal Gaussian distribution is adopted, the above formulation admits a closed-form solution. This symmetric consistency constraint jointly considers the matching of semantic centers and uncertainties, allowing the alignment strength to adaptively adjust according to modality reliability. Summing over all modality pairs yields the modality-level bidirectional consistency alignment loss:

$$L_{bi-align} = \sum_{m<k} L_{bi-align}^{m,k}. \tag{3}$$

### 3.2. Token-level Bidirectional Consistency Alignment

Modality-level alignment ensures global semantic consistency, but different modalities may still exhibit discrepancies in local structures and fine-grained semantic expressions. To further enhance the reliability of cross-modal alignment, we introduce a token-level consistency alignment mechanism based on bidirectional reversible reconstruction.

Let the token representations of modality $m$ and modality $k$ corresponding to sample $i$ be defined as:

$$H_i^{(m)} \in \mathbb{R}^{L_m \times d_t}, H_i^{(k)} \in \mathbb{R}^{L_k \times d_t}, \tag{4}$$

respectively. This token is derived from the local features extracted by the shared semantic feature encoder described

in Section 3.1. Then we construct cross-modal linear reconstruction operators using a deterministic least-squares criterion. Specifically, the linear reconstruction operator that reconstructs modality $m$ from modality $k$ is defined as the closed-form solution to the following matrix regression problem:

$$
\begin{aligned}
R^{m \leftarrow k} &= \arg\min_R \left\| H_i^{(m)} - H_i^{(k)} R \right\|_F^2 \\
&= (H_i^{(k)T} H_i^{(k)} + \lambda I)^{-1} H_i^{(k)T} H_i^{(m)},
\end{aligned}
\tag{5}
$$

where $\lambda > 0$ is a regularization coefficient used to improve numerical stability. In implementation, we use an MLP-based tokenizer to map each modality into a compact token subspace, with both the token length and token dimension set to 8. Thus, Eq. (5) is solved in a low-dimensional space with matched dimensions. With the regularization term, Eq. (5) can be viewed as a ridge-regularized least-squares problem, which guarantees the existence and uniqueness of the closed-form solution. This solution constructs a stable linear reconstruction operator to characterize the optimal linear mapping from one modality to another. If two modalities share consistent local semantic structures, their bidirectional reconstruction operators should also be consistent. Based on this principle, we introduce an operator-level bidirectional consistency constraint:

$$
\begin{aligned}
L_{bi-token} &= \sum_{m<k} \| R^{m \leftarrow k} R^{k \leftarrow m} - I \|_F^2 \\
&+ \sum_{m<k} \| R^{k \leftarrow m} R^{m \leftarrow k} - I \|_F^2,
\end{aligned}
\tag{6}
$$

where $I \in \mathbb{R}^{d_t \times d_t}$ is the identity matrix, corresponding to the ideal mapping that preserves semantics. This constraint is directly imposed on the cross-modal mapping rules themselves, rather than on individual samples or tokens. Therefore, it ensures consistency and reversibility between "viewing modality $k$ from modality $m$" and "viewing modality $m$ from modality $k$".

### 3.3. Multimodal Fusion

In this section, we perform multimodal fusion on the shared semantic features and local token representations and use the fused results for subsequent medical diagnosis classification tasks.

The goal of multimodal fusion is to integrate high-value information that expresses the same medical semantics across different modalities, while suppressing redundancy and noise. To this end, this paper imposes consistency constraints on multimodal representations from multiple conditional perspectives. Specifically, each modality is sequentially treated as a conditional modality, while the remaining modalities serve as complementary modalities. Their provided information is constrained to be consistent

with the semantics of the conditional modality under each conditional perspective. When consistency constraints under all conditional perspectives are satisfied, the multimodal representation is globally self-consistent.

Let the conditional modality of sample $i$ be $p$, and let the set of remaining modalities be $A$. The features of the conditional modality and any remaining modality can then be further decomposed into their respective signal components and noise components:

$$z_i^{(p)} = s_i^{(p)} + n_i^{(p)}, z_i^{(m)} = s_i^{(m)} + n_i^{(m)}, m \in A, \quad (7)$$

where $z_i^{(p)} = [\mu_i^{(p)}; H_i^{(p)}]$ is obtained from the alignment results in Sections 3.1 and 3.2. The signal component $s_i^{(p)}$ is obtained by an MLP, and the noise component $n_i^{(p)}$ is computed by subtraction. We then design a directional information constraint loss to improve semantic consistency between the modality set $A$ and the conditional modality. Specifically, the optimization process is formulated as:

$$L_{signal} = -\sum_{m \in A} (I(s_i^{(p)}; s_i^{(m)} | z_i^{(p)}) - H(s_i^{(m)} | z_i^{(m)})), \quad (8)$$

where $I(\cdot; \cdot)$ denotes conditional mutual information. The mutual information objective is approximated by an InfoNCE-based contrastive loss. It measures the degree to which the signal $s_i^{(m)}$ from set $A$ complements the semantics of the conditional modality. The term $H(\cdot | \cdot)$ represents conditional entropy, which is used to compress the signals from set $A$ and suppress redundant information unrelated to the conditional modality. Meanwhile, we use the following loss function to suppress noise components $n_i^{(p)}$ and $n_i^{(m)}$:

$$L_{noise} = \sum_{m \in A} (\left\| n_i^{(p)} s_i^{(p)T} \right\|^2 + \left\| n_i^{(m)} s_i^{(p)T} \right\|^2)$$
$$+ \sum_{m \in A} (\left\| n_i^{(p)} s_i^{(m)T} \right\|^2 + \left\| n_i^{(m)} s_i^{(m)T} \right\|^2). \quad (9)$$

This term encourages the model to explicitly distinguish signal and noise components during feature extraction. During model training, we minimize the two loss functions above. On this basis, we fuse the signal components of each modality as the final multimodal fusion result for sample $i$:

$$X_{fusion} = \sum_{k=1}^{M} s_i^{(k)}, \quad (10)$$

where $M$ denotes the number of modalities corresponding to sample $i$. $X_{fusion}$ is finally fed into a classifier to perform the multimodal medical classification task. We provide more detailed methodological descriptions and theoretical analyses for Secs. 3.1, 3.2, and 3.3 in Appendices A, B, and C, respectively.

# 4. Experiments

## 4.1. Experiment Settings

**Dataset Setup:** We evaluate the proposed method on five multimodal medical datasets to validate its effectiveness and generalization ability. These datasets include two public brain imaging benchmarks, ABIDE(Di Martino et al., 2014) and ADHD-200(Bellec et al., 2017), and one public breast cancer imaging dataset, CMMD(Cai et al., 2023). In addition, we construct two multimodal datasets for further evaluation. The two self-constructed datasets consist of a breast cancer dataset and a gastric cancer dataset. We provide detailed descriptions of the public datasets, as well as the construction process and content composition of our self-constructed datasets, in Appendix D.

**Experimental Details:** During experiments, the model is trained for 300 epochs, with the initial learning rate set to 1e-4 and the weight decay coefficient set to 5e-4. In addition, an early stopping strategy with a patience of 100 epochs is employed, where training is terminated early if validation performance no longer improves over consecutive epochs. All experiments are implemented using the PyTorch deep learning framework, and parameter optimization is performed with the Adam optimizer. And we set the modality representation dimension $d$ in Sec. 3.1 to 128, while the token sequence lengths $L_k, L_m$ and the token feature dimension $d_t$ in Sec. 3.2 are all set to 8. Experiments are conducted on a computing environment equipped with an NVIDIA GeForce 5090 GPU. In addition, we conduct a sensitivity analysis of the hyperparameter $\lambda$ in Appendix E.

**Evaluation Metrics:** To comprehensively evaluate model performance in medical classification tasks, this paper adopts accuracy (ACC), sensitivity (SEN), specificity (SPE), and the area under the receiver operating characteristic curve (AUC) as evaluation metrics. All metrics are computed based on test-set results from ten-fold cross-validation.

## 4.2. Main Results

To verify the effectiveness of the proposed method in multimodal medical classification tasks, this paper conducts systematic comparisons with various traditional machine learning methods and state-of-the-art multimodal approaches. The compared deep learning methods include Brain-GNN, DGCN(Zhao et al., 2022), AL-NEGAT(Chen et al., 2022), A-GCL(Zhang et al., 2023a), Pop-GCN, GATE(Peng et al., 2022), EV-GCN(Huang & Chung, 2022), MMGL(Zheng et al., 2022), MM-GTUNets(Cai et al., 2025), FOAA(Alwazzan et al., 2024), HetMed(Kim et al., 2023), and Meta(Zhang et al., 2023b), covering both unimodal and multimodal graph learning models.

For the ABIDE and ADHD-200 datasets, the experimental results are shown in Table 1. Unimodal methods based

*Table 1.* Performance comparison of different methods on ABIDE and ADHD-200 datasets. "MM" indicates multi-modal data usage: "×" for single-modal, "✓" for multi-modal. (bold: optimal, underline: suboptimal)

| Method | MM | ABIDE | | | | ADHD-200 | | | |
|---|---|---|---|---|---|---|---|---|---|
| | | ACC (%) | SEN (%) | SPE (%) | AUC (%) | ACC (%) | SEN (%) | SPE (%) | AUC (%) |
| Brain-GNN | × | 69.76 (3.80) | 67.47 (3.10) | 73.28 (3.26) | 72.50 (3.10) | 65.26 (3.60) | 68.59 (3.30) | 63.05 (4.00) | 66.02 (5.50) |
| DGCN | × | 71.83 (2.90) | 70.90 (2.59) | 71.80 (3.10) | 72.45 (2.98) | 68.81 (3.69) | 69.08 (4.07) | 69.53 (4.71) | 70.05 (4.57) |
| AL-NE_GAT | ✓ | 73.17 (3.32) | 78.18 (2.66) | 73.28 (3.14) | 75.02 (2.56) | 69.15 (3.82) | 69.28 (4.32) | 72.26 (3.92) | 68.25 (3.69) |
| A-GCL | × | 79.04 (2.40) | 81.42 (3.03) | 80.95 (3.19) | 82.86 (2.91) | 80.11 (4.30) | **82.04 (4.58)** | 80.08 (4.10) | 78.78 (4.39) |
| Pop-GCN | ✓ | 68.43 (0.86) | 78.32 (0.89) | 57.51 (5.99) | 73.90 (3.08) | 75.45 (0.32) | 50.99 (1.96) | 90.11 (0.22) | 81.72 (1.40) |
| GATE | × | 74.65 (2.50) | 75.59 (2.43) | 76.87 (2.27) | 76.87 (2.27) | 72.20 (0.23) | 77.27 (5.20) | 72.40 (3.78) | 74.61 (3.30) |
| EV-GCN | ✓ | 80.95 (0.27) | 83.74 (0.61) | 77.69 (0.18) | 82.37 (0.29) | 80.95 (0.96) | 60.33 (7.64) | 93.11 (0.13) | 87.15 (1.15) |
| MMGL | ✓ | 80.34 (0.18) | 80.25 (0.21) | 77.16 (3.15) | 79.83 (1.06) | 77.59 (3.64) | 76.21 (2.98) | 74.57 (1.25) | 78.47 (2.29) |
| MM-GTUNets | ✓ | 82.92 (0.54) | 84.22 (0.81) | 81.43 (0.64) | 88.21 (0.61) | 82.68 (0.60) | 77.58 (2.11) | 85.76 (0.46) | 90.71 (0.72) |
| Ours | ✓ | **84.67 (0.32)** | **87.03 (0.56)** | **85.22 (0.24)** | **90.57 (0.69)** | **87.15 (0.12)** | 78.35 (1.83) | **96.17 (0.46)** | **92.29 (0.72)** |

*Table 2.* Performance comparison of different methods on CMMD and Self-constructed Breast datasets. "MM" indicates multi-modal data usage: "×" for single-modal, "✓" for multi-modal. (bold: optimal, underline: suboptimal)

| Method | MM | CMMD | | | | Breast Cancer (Self-constructed) | | | |
|---|---|---|---|---|---|---|---|---|---|
| | | ACC (%) | SEN (%) | SPE (%) | AUC (%) | ACC (%) | SEN (%) | SPE (%) | AUC (%) |
| FOAA | ✓ | 83.42 (1.78) | 81.90 (2.15) | 84.76 (1.96) | 90.13 (1.22) | 63.85 (2.31) | 61.08 (2.88) | 66.27 (2.44) | 71.90 (1.84) |
| DGCN | × | 84.36 (1.66) | 82.74 (2.02) | 85.68 (1.84) | 90.92 (1.18) | 64.92 (2.14) | 61.93 (2.71) | 67.44 (2.33) | 72.74 (1.71) |
| HetMed | ✓ | 85.21 (1.59) | 84.08 (1.88) | 86.14 (1.77) | 91.86 (1.12) | 65.88 (2.06) | 63.02 (2.59) | 68.17 (2.25) | 73.83 (1.64) |
| A-GCL | × | 86.74 (1.42) | 86.05 (1.71) | 87.52 (1.61) | 92.78 (1.00) | 66.97 (1.88) | 64.10 (2.42) | 69.46 (2.12) | 74.92 (1.52) |
| Pop-GCN | ✓ | 85.06 (1.53) | 87.88 (1.47) | 82.41 (2.13) | 92.10 (1.05) | 65.31 (2.20) | 68.42 (2.10) | 62.18 (3.01) | 73.26 (1.78) |
| GATE | × | 85.92 (1.46) | 85.31 (1.78) | 86.63 (1.66) | 92.34 (1.03) | 66.42 (1.96) | 64.25 (2.41) | 68.68 (2.19) | 74.38 (1.56) |
| Meta | ✓ | 88.94 (1.12) | 88.31 (1.36) | 89.52 (1.25) | 93.84 (0.86) | 68.12 (1.60) | 65.74 (2.18) | 70.45 (1.94) | 75.72 (1.31) |
| MMGL | ✓ | 87.83 (1.20) | 87.14 (1.45) | 88.46 (1.32) | 93.11 (0.92) | 67.54 (1.72) | 64.98 (2.24) | 69.92 (2.01) | 75.04 (1.38) |
| MM-GTUNets | ✓ | 88.56 (1.10) | 87.92 (1.41) | 89.18 (1.28) | 93.62 (0.88) | 67.86 (1.64) | 65.33 (2.20) | 70.12 (1.98) | 75.38 (1.34) |
| Ours | ✓ | **92.37 (0.86)** | **91.84 (1.06)** | **92.89 (0.98)** | **97.12 (0.63)** | **71.64 (1.42)** | **69.10 (2.06)** | **73.92 (1.88)** | **79.45 (1.17)** |

*Table 3.* Performance comparison of different methods on the self-constructed Gastric Cancer dataset. "MM" indicates multi-modal data usage: "×" for single-modal, "✓" for multi-modal. (bold: optimal, underline: suboptimal)

| Method | MM | Gastric Cancer (Self-constructed) | | | |
|---|---|---|---|---|---|
| | | ACC (%) | SEN (%) | SPE (%) | AUC (%) |
| FOAA | ✓ | 66.12 (2.61) | 64.05 (3.04) | 68.02 (2.70) | 73.21 (1.96) |
| DGCN | × | 66.94 (2.42) | 65.16 (2.95) | 68.63 (2.58) | 74.05 (1.86) |
| HetMed | ✓ | 67.88 (2.33) | 66.74 (2.81) | 68.92 (2.53) | 75.26 (1.71) |
| A-GCL | × | 68.35 (2.14) | 66.89 (2.63) | 69.63 (2.36) | 76.11 (1.60) |
| Pop-GCN | ✓ | 67.42 (2.29) | 71.08 (2.08) | 64.11 (3.02) | 75.74 (1.66) |
| GATE | × | 68.06 (2.18) | 66.20 (2.72) | 69.75 (2.40) | 75.88 (1.63) |
| Meta | ✓ | 69.02 (1.51) | 67.55 (2.12) | 70.47 (1.91) | 78.02 (1.20) |
| MMGL | ✓ | 68.74 (1.62) | 67.05 (2.20) | 69.90 (1.98) | 77.41 (1.27) |
| MM-GTUNets | ✓ | 68.91 (1.57) | 67.28 (2.16) | 70.13 (1.95) | 77.88 (1.22) |
| Ours | ✓ | **73.11 (1.36)** | **71.08 (2.04)** | **74.88 (1.71)** | **81.26 (1.12)** |

on graph neural networks, such as Brain-GNN and DGCN, show a clear performance decline. Due to reliance on single-modality modeling, their classification performance has an inherent upper bound. Overall, multimodal methods outperform unimodal methods, validating the effectiveness of multimodal information fusion in brain disease prediction tasks. Among them, EV-GCN and MMGL achieve relatively

stable performance across multiple metrics, with particularly strong results on AUC. On the ABIDE dataset, our method achieves the highest values on ACC, SEN, SPE, and AUC, especially achieving 90.57% on AUC, which indicates stronger overall discriminative ability across different thresholds. On the ADHD-200 dataset, our method also achieves the best results on ACC and AUC, while maintaining a good balance between SEN and SPE. These results demonstrate that, through modality-level and token-level consistency modeling, the proposed method effectively alleviates modality discrepancies and noise interference in multimodal brain imaging data.

The experimental results on the CMMD dataset and self-constructed breast cancer multimodal dataset are shown in Table 2. The proposed method also achieves competitive results in breast cancer molecular subtyping tasks. Unlike brain imaging data, breast cancer multimodal data exhibit significant imbalance in modality scale, information granularity, and class distribution, with different modalities exhibiting large differences in discriminative contributions across subtypes. Comparative experimental results show that some methods are sensitive to differences in cross-

*Table 4.* Performance comparison of different methods under increasing modality numbers on self-constructed Breast and Gastric datasets. "MS" indicates different modality settings. (bold: optimal)

| Method | MS | Breast Cancer (Self-constructed) | | | | Gastric Cancer (Self-constructed) | | | |
|---|---|---|---|---|---|---|---|---|---|
| | | ACC (%) | SEN (%) | SPE (%) | AUC (%) | ACC (%) | SEN (%) | SPE (%) | AUC (%) |
| Meta | $\{1, 2\}$ | 68.12 (1.60) | 65.74 (2.18) | 70.45 (1.94) | 75.72 (1.31) | 69.02 (1.51) | 67.55 (2.12) | 70.47 (1.91) | 78.02 (1.20) |
| Ours | $\{1, 2\}$ | 71.64 (1.42) | 69.10 (2.06) | 73.92 (1.88) | 79.45 (1.17) | 73.11 (1.36) | 71.08 (2.04) | 74.88 (1.71) | 81.26 (1.12) |
| Meta | $\{1, 2, 3\}$ | 72.61 (1.13) | 71.38 (1.07) | 74.55 (1.35) | 80.76 (1.41) | 73.56 (1.22) | 71.87 (0.94) | 74.57 (1.65) | 81.94 (1.37) |
| Ours | $\{1, 2, 3\}$ | 73.92 (1.18) | 72.35 (1.97) | 75.21 (1.63) | 82.04 (1.06) | **74.62 (1.21)** | **72.80 (1.96)** | **75.98 (1.58)** | **82.73 (1.03)** |
| Meta | $\{1, 2, 3, 4\}$ | 74.03 (0.98) | 72.89 (1.72) | 75.34 (1.67) | 82.61 (1.16) | | | - | |
| Ours | $\{1, 2, 3, 4\}$ | **75.38 (1.05)** | **74.02 (1.82)** | **76.64 (1.49)** | **83.97 (0.94)** | | | - | |

*Table 5.* Performance comparison under missing-modality settings. (bold: optimal)

| Method | Modality Count | Missing Count | Breast Cancer (Self-constructed) | | | | Gastric Cancer (Self-constructed) | | | |
|---|---|---|---|---|---|---|---|---|---|---|
| | | | ACC (%) | SEN (%) | SPE (%) | AUC (%) | ACC (%) | SEN (%) | SPE (%) | AUC (%) |
| Meta | 3 | 1 | 70.94 (1.26) | 69.49 (1.74) | 71.95 (1.34) | 79.12 (1.09) | 72.18 (1.32) | 70.82 (0.86) | 72.37 (1.05) | 80.56 (1.29) |
| Ours | 3 | 1 | 72.31 (0.87) | 70.96 (1.09) | 73.24 (1.37) | 80.83 (1.61) | **73.46 (1.72)** | **72.06 (0.89)** | **73.68 (1.29)** | **81.67 (1.42)** |
| Meta | 4 | 1 | 72.86 (0.98) | 71.48 (1.38) | 73.91 (1.28) | 81.37 (1.53) | | | - | |
| Ours | 4 | 1 | **74.12 (1.11)** | **72.87 (1.57)** | **75.07 (0.97)** | **82.58 (1.14)** | | | - | |

modal information density or class imbalance, leading to notable fluctuations, especially in sensitivity and specificity. In contrast, the proposed method maintains stable performance on core metrics such as ACC and AUC on both datasets. This indicates that the proposed multi-level consistency alignment and fusion mechanism effectively integrates complementary discriminative information across modalities.

As shown in Table 3, experiments on the self-constructed gastric cancer multimodal dataset further validate the generalization ability of the proposed method. This dataset exhibits more pronounced differences across modalities, and the task involves five-class classification, which increases modeling difficulty. Comparative results show that some methods suffer notable performance degradation when modality structures change, whereas the proposed method maintains strong classification performance. These findings indicate that the proposed fusion strategy does not rely on fixed modality structures and demonstrates good adaptability in cross-disease and multimodal scenarios.

To further validate the adaptability and scalability of the proposed method to variations in the number of modalities, we design experiments with an increasing number of modalities. We select the self-constructed breast cancer dataset, which includes four modalities, and the self-constructed gastric cancer dataset, which includes three modalities, as evaluation benchmarks. For the breast cancer dataset, we construct modality combinations 1,2, 1,2,3, and 1,2,3,4. For the gastric cancer dataset, we construct modality combinations 1,2 and 1,2,3. The experimental results are shown in Table 4.

Experimental results show that both the adapted Meta-Transformer and the proposed method run stably and

achieve reasonable performance under different modality number settings, demonstrating strong adaptability to variations in modality number. More importantly, as the number of available modalities increases, all models' performance exhibits an upward trend, indicating that multimodal information is highly complementary. At the same time, the results show that the proposed hierarchical consistency alignment and fusion mechanism can effectively absorb incremental information from newly introduced modalities. And compared with Meta-Transformer, the proposed method achieves consistent improvements under different modality settings.

To further evaluate the robustness of the proposed method under varying modality availability, we additionally conduct a missing-modality experiment. This experiment simulates incomplete modality inputs by randomly removing part of the available modalities. The model is compared with Meta-Transformer under the same missing-modality settings. As shown in Table 5, the proposed method consistently outperforms Meta-Transformer under different missing-modality settings. When one modality is randomly removed, this method still maintains higher ACC and AUC on both the Breast Cancer and Gastric Cancer datasets. This robustness mainly benefits from the modality-agnostic fusion design and the multi-view consistency constraint, which allow the model to construct a stable multimodal representation from the currently available modalities.

In addition, to examine whether a more complex final aggregation module is necessary after consistency optimization, we compare three final aggregation strategies, including MLP-based fusion, cross-attention-based fusion and ours. And for the self-constructed Breast Cancer dataset, we select ultrasound image and clinical text as the two input

*Table 6.* Performance comparison of final aggregation module on CMMD and Self-constructed Breast datasets. (bold: optimal)

| Setting | CMMD | | | | Breast Cancer (Self-constructed) | | | |
|---|---|---|---|---|---|---|---|---|
| | ACC (%) | SEN (%) | SPE (%) | AUC (%) | ACC (%) | SEN (%) | SPE (%) | AUC (%) |
| MLP-based fusion | 91.86 (1.12) | 91.36 (1.53) | 91.19 (1.32) | 96.68 (0.95) | 70.52 (1.60) | 68.33 (1.84) | 72.62 (2.03) | 78.62 (1.28) |
| Cross-attention-based fusion | 92.01 (2.13) | 91.78 (2.06) | 92.21 (1.37) | 96.84 (0.98) | 70.88 (1.42) | 68.56 (1.29) | 73.06 (2.10) | 78.95 (0.83) |
| Ours | **92.37 (0.86)** | **91.84 (1.06)** | **92.89 (0.98)** | **97.12 (0.63)** | **71.64 (1.42)** | **69.10 (2.06)** | **73.92 (1.88)** | **79.45 (1.17)** |

*Table 7.* Performance comparison of linear reconstruction strategy on CMMD and Self-constructed Breast datasets. (bold: optimal)

| Setting | CMMD | | | | Breast Cancer (Self-constructed) | | | |
|---|---|---|---|---|---|---|---|---|
| | ACC (%) | SEN (%) | SPE (%) | AUC (%) | ACC (%) | SEN (%) | SPE (%) | AUC (%) |
| MLP-based nonlinear mapping | 91.68 (1.34) | 91.57 (0.85) | 92.64 (1.18) | 96.54 (1.71) | 70.93 (2.54) | 68.62 (1.60) | 73.36 (0.97) | 79.26 (1.32) |
| Ours | **92.37 (0.86)** | **91.84 (1.06)** | **92.89 (0.98)** | **97.12 (0.63)** | **71.64 (1.42)** | **69.10 (2.06)** | **73.92 (1.88)** | **79.45 (1.17)** |

modalities. The experimental results are shown in Table 6.

The results show that MLP and cross-attention model inter-modal interactions by introducing additional parameters, but their overall classification performance is slightly lower than that of the proposed method. This further demonstrates that multi-view consistency modeling has already effectively extracted high-quality shared signals. The current fusion strategy can more stably integrate useful information without requiring additional complex parameterized aggregators.

In the proposed method, the role of bidirectional linear reconstruction is to examine whether local semantic structures can mutually explain each other, thereby providing an interpretable constraint signal for token-level consistency. More specifically, this method relies on a local linear approximation, where the aligned semantic structures already provide a basis for similarity analysis. To further validate this point, we replaced the linear reconstruction with an MLP-based nonlinear mapping for comparison. The experimental results are shown in Table 7.

The experimental results show that the MLP-based variant performs almost on par with our method, and in some cases even leads to lower classification accuracy. These results further support that our method effectively captures local linear approximation rather than relying on a global linear assumption.

### 4.3. Ablation Experiments

**Analysis of the Effect of Uncertainty Modeling:** To verify the effectiveness of modality-level semantic uncertainty modeling in the proposed method, we remove the Gaussian-based semantic uncertainty modeling mechanism from the model and retain only deterministic shared semantic representations for cross-modal alignment. To further intuitively analyze the impact of uncertainty modeling on multimodal fused representations, we perform t-SNE visualization of the fused features produced by the full model and the ablation model. The CMMD dataset is used, and the results

are shown in Figure 2. From the visualization results, it can be observed that without uncertainty modeling, samples from different classes exhibit more pronounced overlap in the low-dimensional embedding space, and intra-class compactness is weaker. In contrast, after introducing semantic uncertainty modeling, the fused features show more compact intra-class distributions and clearer inter-class boundaries in the embedding space. This phenomenon indicates that uncertainty modeling enhances the discriminability of multimodal features at the representation level.

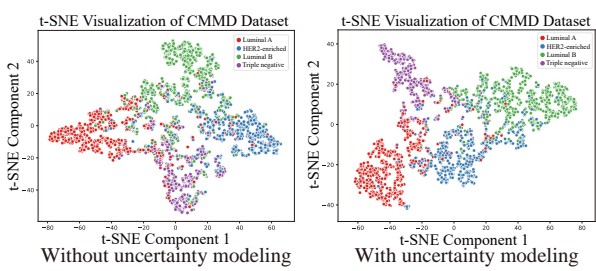

*Figure 2.* t-SNE Visualization Analysis of Uncertainty Modeling.

**Analysis of the Effect of Token-Level Consistency Constraints:** To analyze the role of the proposed token-level bidirectional consistency constraint in fine-grained cross-modal semantic modeling, we replace the original token-level consistency modeling based on deterministic linear reconstruction and reversible closed-loop constraints with a cosine-similarity-based local feature alignment scheme. We visualize channel activations output by the shared semantic feature encoder and compare the Top-8 most highly activated channels under each setting. We use our self-constructed breast cancer dataset, and the visualization results are shown in Figure 3. In the ablation model that uses cosine similarity for token alignment, highly activated channels from different modalities show large differences in spatial distribution and response patterns. And some channels respond strongly to non-discriminative regions. In contrast, in the full model, after introducing token-level structural consistency constraints, highly activated channels

exhibit more consistent response patterns across modalities. And their attention regions are more concentrated on key structural areas relevant to the classification task. These observations indicate that the proposed token-level consistency modeling not only improves numerical performance, but also enhances semantic consistency and interpretability of cross-modal representations at the feature level.

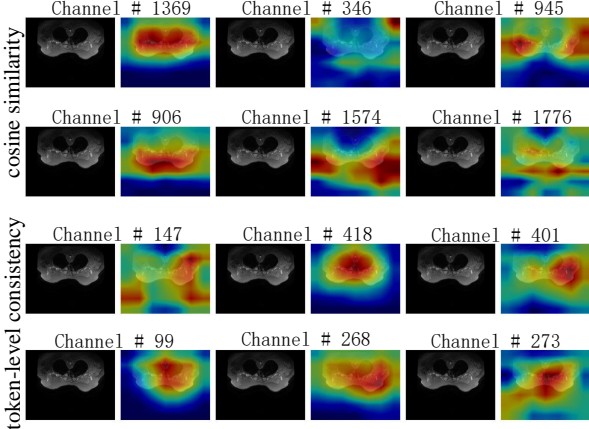

*Figure 3.* Channel Activation Maps of the Shared Semantic Feature Encoder Outputs Under Different Ablation Settings.

**Fusion Strategy Analysis:** To systematically analyze the effectiveness of key components in the proposed fusion module, we conduct ablation experiments on the self-constructed breast cancer multimodal dataset. The experiments focus on the impact of the multi-view fusion mechanism and the signal–noise decomposition with noise suppression strategy on classification performance. We design four comparative settings to clearly reveal the individual contributions of each module. The first setting is the full model (Ours), which retains both the multi-view fusion mechanism and signal–noise decomposition with suppression; the second setting removes multi-view fusion (w/o Multi-view) by replacing conditional-view-based fusion with direct concatenation; the third setting removes signal–noise decomposition and denoising (w/o Denoise) and directly fuses the original modality features; the fourth setting removes both mechanisms (w/o Both), adopting one-shot symmetric fusion without signal–noise decomposition or noise suppression.

We plot per-class ROC curves for the four experimental settings. The visualization results are shown in Figure 4. The full model achieves ROC curves closer to the upper-left corner for most classes, indicating higher true positive rates and lower false positive rates. In contrast, the ROC curves of w/o Denoise and w/o Both show a more pronounced downward shift. This indicates that classification capability degrades when noise is not effectively suppressed, especially for more easily confused classes. Furthermore, the ROC curves of w/o Multi-view show increased threshold sensitivity for some classes, with larger performance fluctu-

ations across different thresholds. This reveals that without multi-view fusion, the model struggles to fully integrate complementary multimodal information, which affects the stability of inter-class boundaries. Overall, the class-wise ROC curves validate the necessity of combining multi-view consistency fusion with signal-noise suppression in the proposed fusion module. The former improves the effectiveness of multimodal information integration, while the latter reduces noise-induced misclassification.

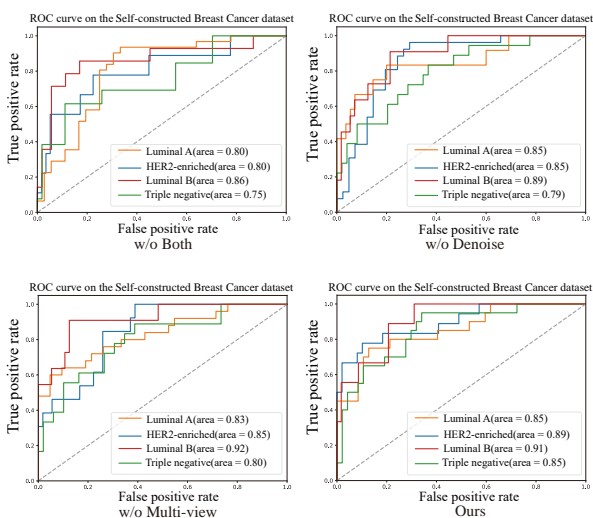

*Figure 4.* ROC Curves Under Different Fusion Strategies.

In addition, Appendix F provides more detailed technical-level quantitative ablation studies.

## 5. Conclusion

This paper addresses challenges in medical multimodal classification, including unstable semantic alignment, limited scalability of fusion strategies, and proposes a hierarchical consistency alignment and fusion framework. Specifically, semantic uncertainty modeling is introduced at the modality level to enable reliable cross-modal alignment. At the token level, a bidirectional consistency constraint based on deterministic linear reconstruction is proposed to characterize fine-grained semantic mutual verification. In addition, a multi-view fusion mechanism is designed to remove noise interference while mining shared semantics across modalities, allowing the fusion module to naturally scale to an arbitrary number of modality inputs. Future work will focus on three directions. First, we will further investigate robust inference under missing modalities and dynamic modality combinations. Second, we will extend the framework to more complex clinical tasks, such as segmentation and prognosis prediction. Third, we will conduct validation on larger multi-center clinical datasets to enhance generalization ability and practical clinical applicability.

## Acknowledgements

This work was supported by the Special Funds of Taishan Scholars Project of Shandong Province (tstp20221137), Key R&D Program of Shandong Province (Competitive Innovation Platform) Project (2025CXPT100), Key R&D Program of Shandong Province (Major Science and Technology Innovation Project) (2025CXGC020101), and Jinan Talent Development Project-Science and Technology (202333037).

## Impact Statement

This paper proposes a modality-agnostic multimodal fusion framework for medical diagnosis, aiming to improve the robustness and scalability of multimodal learning under varying modality availability. By integrating uncertainty-aware modality-level alignment, bidirectional token-level consistency constraints, and a multi-view fusion strategy, the proposed method enables more reliable representation learning when multimodal signals are heterogeneous, noisy, or partially missing. This capability may positively impact real-world clinical decision support systems, where the number and quality of available modalities often differ across patients and institutions.

From a broader perspective, the proposed framework may help reduce performance degradation caused by modality imbalance or inconsistent acquisition protocols, potentially improving the accessibility and generalization of multimodal medical AI models in practical deployment. Nevertheless, limitations remain. The method may inherit biases from the training data, including demographic and site-specific biases, which could lead to uneven performance across patient populations. Moreover, clinical adoption requires careful validation, model calibration, and transparency, since incorrect predictions in medical settings can lead to harmful consequences. We encourage future work to evaluate fairness across subgroups, study uncertainty calibration under distribution shift, and collaborate with clinical experts to ensure safe and responsible deployment.

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

## A. Analysis of the Effectiveness of Semantic Alignment in Sec. 3.1

This section explains and theoretically justifies that the modality-level consistency learning in Sec. 3.1 can achieve cross-modal alignment at the semantic level. It should be emphasized that the semantic alignment implemented in this work does not refer to simply bringing feature vectors from different modalities closer in Euclidean space. Instead, it means that, for the same sample, different modalities should provide consistent interpretations of the same latent medical semantics. That is, they should form unified and fusion-ready semantic representations in a shared semantic space. Medical multimodal data exhibit pronounced heterogeneity and noise discrepancies, and different modalities (such as MRI, ultrasound, pathology, and text) inherently differ in the reliability with which they express the same disease semantics. If alignment is directly performed using deterministic vectors, noisy modalities often interfere with the construction of the shared space. Therefore, the key design of Sec. 3.1 lies in elevating each modality's semantic characterization from point estimation to distribution estimation. This allows the model to simultaneously express semantic content and semantic confidence, thereby endowing the alignment objective with stronger robustness and interpretability.

Specifically, for the $m$-th modality input $z_i^{(m)} \in \mathbb{R}^d$ of sample $i$, we introduce a latent semantic variable $s_i$ to represent the disease semantics of this sample in the shared space. Each modality outputs its estimated mean and uncertainty characterization of this semantics through a semantic encoder and an uncertainty encoder, respectively:

$$\mu_i^{(m)} = f_\mu^{(m)}(z_i^{(m)}), \sigma_i^{(m)} = f_\sigma^{(m)}(z_i^{(m)}). \tag{11}$$

Here, $\sigma_i^{(m)}$ denotes the element-wise standard deviation, which provides an inverse characterization of confidence. Based on this, Sec. 3.1 represents the estimation of $s_i$ by modality $m$ as a diagonal Gaussian distribution:

$$q_i^{(m)}(s_i) = \mathrm{N}(s_i; \mu_i^{(m)}, diag((\sigma_i^{(m)})^2)). \tag{12}$$

The significance of this distribution-based semantic modeling lies in the following aspect. The target of modality-level semantic alignment is no longer a deterministic vector $\mu$, but a complete interpretation that includes both semantic content and semantic reliability. As a result, semantic alignment requires not only consistency of semantic meanings across modalities, but also consistency in their confidence regarding semantic judgments.

Under distribution-based modeling, the key to achieving semantic alignment lies in selecting an objective function that can measure distributional differences and strictly characterize consistency. Sec. 3.1 adopts symmetric KL divergence as the consistency constraint:

$$SKL(q_i^{(m)}, q_i^{(k)}) = KL(q_i^{(m)} \| q_i^{(k)}) + KL(q_i^{(k)} \| q_i^{(m)}). \tag{13}$$

And the modality-level alignment loss is obtained by summing over all modality pairs. The core reason for choosing symmetric KL divergence is that it provides a very strong criterion for semantic consistency. KL divergence is non-negative and equals zero if and only if the two distributions are identical, i.e., $KL(p\|q) = 0 \Leftrightarrow p = q$ (almost everywhere). This implies that when optimization drives the KL-based loss toward zero, the model is not merely making vectors more similar. Instead, it forces all modalities to converge to identical distributional interpretations of the same latent semantic variable. The optimum of Sec. 3.1 is strictly equivalent to semantic distributional consistency. Therefore, semantic alignment is guaranteed at the objective-function level, and spurious solutions with minimal loss but inconsistent semantics cannot occur.

To further illustrate that this alignment operates at the semantic level rather than merely at the representation level, we can explain its underlying mechanism using the closed-form expression of diagonal Gaussians. For two diagonal Gaussians $q_a = \mathrm{N}(\mu_a, diag(\sigma_a^2))$ and $q_b = \mathrm{N}(\mu_b, diag(\sigma_b^2))$, their symmetric KL divergence can be written in the following analytic form:

$$SKL(q_a, q_b) = \frac{1}{2} \sum_{j=1}^{d} \left[ (\mu_{a,j} - \mu_{b,j})^2 (\frac{1}{\sigma_{a,j}^2} + \frac{1}{\sigma_{b,j}^2}) + (\frac{\sigma_{a,j}^2}{\sigma_{b,j}^2} + \frac{\sigma_{b,j}^2}{\sigma_{a,j}^2} - 2) \right]. \tag{14}$$

This expression directly reveals what Sec. 3.1 aligns during optimization. The first term penalizes the squared difference of means, thereby promoting consistency of semantic content. This penalty is not the simple term $\|\mu_a - \mu_b\|^2$, but is adaptively weighted by the precision term $1/\sigma^2$. As a result, reliable modalities exert stronger alignment effects, while unreliable modalities contribute weaker alignment effects. The second term characterizes consistency in variance scale. This term becomes zero only when $\sigma_a = \sigma_b$, thereby encouraging different modalities to converge toward consistent characterizations of semantic reliability.

This effect can be further explained from a gradient perspective to analyze the alignment dynamics. Taking the derivative with respect to $\mu_a$, we obtain:

$$\frac{\partial}{\partial_{\mu_{a,j}}} SKL(q_a, q_b) = \sum_{j=1}^{d} \left[ (\mu_{a,j} - \mu_{b,j})(\frac{1}{\sigma_{a,j}^2} + \frac{1}{\sigma_{b,j}^2}) \right]. \tag{15}$$

This gradient form indicates that the update strength of semantic mean alignment is proportional to the precision $1/\sigma^2$, thus exhibiting a clear reliability-adaptive property. If a modality is unreliable on a certain semantic dimension (i.e., large $\sigma$), the corresponding precision is small and the alignment gradient is weak, so it does not forcibly pull the shared space. In contrast, reliable modalities more strongly drive the formation of consistency. Especially in real clinical data, where noisy or missing modalities are common, this mechanism explains why Sec. 3.1 can achieve robust semantic alignment without introducing manually specified weights.

## B. Theoretical Discussion and Effectiveness Analysis of Token-Level Bidirectional Consistency in Sec. 3.2

Sec. 3.1 achieves cross-modal global semantic alignment from the perspective of modality-level semantic distribution consistency, but global semantic consistency alone is insufficient to support high-quality fusion. The reason is that complementarity in medical multimodal data is often manifested through local structural differences. Examples include tumor boundary textures, local focal regions, and textual descriptions of specific lesions, which cannot be fully captured by a single global feature. More importantly, different modalities exhibit inherent structural heterogeneity at the local level. The local token expressions of the same semantics across modalities are not token-wise aligned, but are closer to subspace-level consistency or locally interpretable structures. Therefore, the goal of Sec. 3.2 is to build upon Sec. 3.1 by further constraining different modalities to maintain consistency in local token representations. This enables the formation of truly fusion-ready semantic structures at a fine-grained level.

To this end, Sec. 3.2 does not adopt conventional token similarity alignment methods, such as cosine similarity or cross-attention weights. Instead, it introduces a more structured and interpretable bidirectional linear reconstruction consistency constraint. Specifically, let the token representations of modality $m$ and modality $k$ for sample $i$ be denoted as:

$$H_i^{(m)} \in \mathbb{R}^{L_m \times d_t}, H_i^{(k)} \in \mathbb{R}^{L_k \times d_t}. \tag{16}$$

Each row corresponds to the representation of a local token, and these tokens are derived from the local outputs of the shared semantic encoder in Sec. 3.1. This point is critical. If tokens lack a global semantic foundation, imposing structural constraints at the token level may forcibly align noise structures across modalities. After global semantics have been aligned in Sec. 3.1, the objective of token-level constraints becomes much clearer.

The core idea of Sec. 3.2 can be summarized in one sentence. If local tokens of modality $k$ can explain local tokens of modality $m$, this explanation rule should also hold in the reverse direction, and the two explanations should be mutually inverse mappings. Mathematically, such consistency should not be reflected only in token similarity, but in the reversibility of the explanation rule itself. Therefore, we reformulate cross-modal token alignment as a deterministic matrix regression problem. Specifically, we learn a linear operator such that tokens of modality $k$ can linearly reconstruct tokens of modality $m$.

In particular, the linear operator that reconstructs modality $m$ from modality $k$ is defined as the closed-form solution of the following regularized least-squares problem:

$$R^{m \leftarrow k} = \arg \min_R \left\| H_i^{(m)} - H_i^{(k)} R \right\|_F^2 = (H_i^{(k)T} H_i^{(k)} + \lambda I)^{-1} H_i^{(k)T} H_i^{(m)}. \tag{17}$$

From a theoretical perspective, this operator provides an optimal linear rule for explaining the token subspace of modality $m$ using the token subspace of modality $k$. In other words, Sec. 3.2 does not require one-to-one correspondence between tokens. Instead, it requires an interpretable linear transformation between their local semantic structures. This assumption better matches practical medical multimodal scenarios. Local expressions from different modalities may differ in space or form. As long as they describe the same semantic structure, a stable explanation rule should exist.

Importantly, a single-direction linear reconstruction is insufficient to guarantee alignment. Even if $H^{(k)}R \approx H^{(m)}$, projection degeneration or information loss may still occur. For example, $R$ may map tokens of modality $k$ to a low-dimensional subspace, fitting part of the structure of modality $m$ while ignoring critical differences. In this case, $R$

behaves more like a one-way compressor rather than evidence of semantic consistency. Therefore, Sec. 3.2 introduces a reconstruction operator in the reverse direction:

$$R^{k \leftarrow m} = \arg \min_{R} \left\| H_i^{(k)} - H_i^{(m)} R \right\|_F^2. \tag{18}$$

If two modalities are truly consistent in local structure, the explanation rule from $k$ to $m$ and the explanation rule from $m$ to $k$ should match each other and ideally form inverse mappings. Accordingly, the main text introduces an operator-level bidirectional consistency constraint:

$$L_{bi-token} = \sum_{m < k} (\left\| R^{m \leftarrow k} R^{k \leftarrow m} - I \right\|_F^2 + \left\| R^{k \leftarrow m} R^{m \leftarrow k} - I \right\|_F^2). \tag{19}$$

The key advantage of this constraint is that it does not enforce pointwise alignment on token similarity. Instead, it imposes structural constraints on the explanation rules themselves, thereby enabling a stronger mechanism of semantic mutual verification. From a theoretical perspective, bidirectional reversible consistency constraints can guarantee the sufficiency of token-level semantic alignment. This can be understood from two aspects:

- From the perspective of information preservation, if $R^{m \leftarrow k} R^{k \leftarrow m} \approx I$, this implies that $R^{m \leftarrow k}$ is approximately invertible in the semantic subspace. When the constraint converges, it can be assumed that collapse does not occur in the principal shared semantic subspace. Once low-rank collapse happens, it is impossible for a reverse operator to restore the product to the identity mapping.

- From the perspective of structural consistency, this constraint effectively forces cross-modal token representations to satisfy a consistent local geometric structure. If token representations are viewed as point clouds lying in a local semantic subspace, the linear operator $R^{m \leftarrow k}$ constructs a deterministic linear correspondence between the two point clouds. The bidirectional reversible constraint further requires this correspondence to be stable and closed-loop consistent.

## C. Theoretical Discussion and Effectiveness Analysis of Multi-View Consistent Fusion in Sec. 3.3

In Sec. 3.1 and Sec. 3.2, cross-modal consistency has been established at the modality level and the token level, respectively. This provides different modalities with a common semantic foundation for fusion at both global semantics and local structures. However, alignment only ensures that semantic expressions across modalities are comparable and mutually verifiable. Fusion aims to extract truly valuable complementary information from multimodal representations. At the same time, it suppresses redundancy and noise to obtain a unified representation that is most discriminative for diagnosis. In real clinical scenarios, different modalities exhibit significant differences in noise levels and information density. If fusion relies only on one-shot concatenation or simple attention weighting, the fused representation is often dominated by modalities with higher information content. It may also be disturbed by noisy modalities. As a result, the fused representation becomes unstable. Therefore, the essential goal of Sec. 3.3 is not merely to merge multiple modalities. Instead, it is to construct a fusion mechanism that remains self-consistent, robust, and interpretable under an arbitrary number of modalities.

In terms of representation, the fusion inputs of Sec. 3.3 are obtained from all outputs of the shared semantic encoder and the token alignment module in the previous section. Specifically, for the $p$-th conditional modality, we concatenate its modality-level shared semantic representation $\mu_i^{(p)}$ with its token representation $H_i^{(p)}$ to form the fused feature:

$$z_i^{(p)} = [\mu_i^{(p)}; H_i^{(p)}]. \tag{20}$$

It is then used as the semantic reference for the conditional view. Meanwhile, the remaining modalities are denoted as set $A$. They are not treated on equal footing with the conditional modality, but serve as complementary information sources. Under the constraint of the conditional semantics, they are required to provide incremental semantic information rather than alter the primary semantics or introduce noise interference. This design directly addresses a key issue in medical multimodal fusion: auxiliary modalities should play a supportive role, rather than competing with the primary modality for explanatory dominance.

To instantiate the above idea as an optimizable mathematical formulation, Sec. 3.3 first decomposes the fused features of each modality. Let the conditional modality be $p$, and let any complementary modality be denoted as $m \in A$. The

corresponding decomposition is given as:

$$z_i^{(p)} = s_i^{(p)} + n_i^{(p)}, z_i^{(m)} = s_i^{(m)} + n_i^{(m)}, m \in A. \tag{21}$$

This step elevates the fusion problem from adjusting weights on raw features to purifying information at the semantic level, thereby providing a clear optimization objective for subsequent fusion. Therefore, this decomposition is not merely a structural design, but an interpretable modeling of the fusion task. Fusion is not arbitrary aggregation. Instead, it integrates only the components from each modality that belong to shared diagnostic semantics. After the decomposition, to ensure that the signals from complementary modalities truly provide semantic supplementation to the conditional modality rather than noise perturbations, Sec. 3.3 introduces a directional consistency constraint from an information-theoretic perspective. Its core formulation is as follows:

$$L_{signal} = - \sum_{m \in A} (I(s_i^{(p)}; s_i^{(m)} | z_i^{(p)}) - H(s_i^{(m)} | z_i^{(m)})). \tag{22}$$

The theoretical meaning of this formulation can be understood from two perspectives: complementarity and compression. The first term is conditional mutual information. It measures how much incremental information the signal component of a complementary modality can provide that is consistent with the conditional semantics, given the conditional modality semantics. In other words, it encourages complementary modalities to contribute components that remain informative under the conditional semantic interpretation. This ensures that fusion fundamentally follows the design principle that auxiliary modalities serve the primary view. The second term is a conditional entropy term. It imposes compression on the signals of complementary modalities. This removes redundant information that is irrelevant to the conditional semantics and prevents auxiliary modalities from introducing excessive noise into the fusion process. This formulation mathematically defines the role of auxiliary modalities as providing signal components that are consistent with the conditional semantics and have higher information density.

However, constraining signal validity alone is still insufficient to guarantee robust fusion. If noise components become entangled with signal components during fusion, the final fused representation may still be biased by noise, even when the signal components exhibit certain consistency. Therefore, Sec. 3.3 further explicitly suppresses the correlation between noise components and signal components:

$$L_{signal} = \sum_{m \in A} (\left\| n_i^{(p)} s_i^{(p)T} \right\|^2 + \left\| n_i^{(m)} s_i^{(p)T} \right\|^2 + \left\| n_i^{(p)} s_i^{(m)T} \right\|^2 + \left\| n_i^{(m)} s_i^{(m)T} \right\|^2). \tag{23}$$

This formulation encourages noise components to carry as little discriminative information related to the signal as possible in the semantic space. When this term is minimized, the noise components become statistically orthogonal to the semantic signals. This reduces noise interference with the classification boundary. As a result, the fusion process becomes more stable and better reflects the characteristics of real clinical data. This ensures that semantic structures truly useful for diagnosis remain consistent across modalities.

In addition, a key innovation of Sec. 3.3 lies in multi-view consistency. At each step, one modality is selected as the conditional view $p$, while the remaining modalities form the complementary set $A$. The above consistency constraints are then applied under this conditional view. When this process is executed for all conditional views, the model effectively constructs a globally self-consistent fusion mechanism. If a modality produces information that is inconsistent with the conditional semantics under a certain view, it will be suppressed during fusion. When consistency constraints are satisfied under all views, the resulting fused representation no longer depends on the preference of any single modality. Instead, it forms a stable cross-modal consensus under multi-view constraints. During fusion, this principle is generalized such that the constraint should hold when viewing all other modalities from any given modality. This structurally guarantees that the fused representation is not subordinate to any single modality, but a stable semantic outcome jointly constrained by all modalities.

Finally, from the perspective of scalability, the fusion mechanism in Sec. 3.3 naturally supports an arbitrary number of modality inputs. This is because modalities are treated as a set during fusion. The process depends only on the currently available modality set and does not require a fixed number of modalities. It also does not require redesigning network structures or loss functions for different modality combinations. When a new modality appears, it only needs to be included in set $A$ or treated as a new conditional view $p$. When a modality is missing, it simply does not participate in fusion or consistency constraints, and the model can still operate normally. Therefore, this fusion strategy achieves true

modality-number agnosticism. Both the network architecture and the optimization objective are unbiased with respect to the number of modalities. Moreover, because multi-view consistency enforces that newly added modalities must agree with the consensus formed by existing modalities, new modalities do not introduce training instability or performance degradation. Instead, they can improve fusion quality by providing additional complementary signals.

## D. Detailed Dataset Description

To comprehensively evaluate the effectiveness and generalization ability of the proposed method in medical multimodal diagnostic tasks, we conduct experiments on five medical multimodal datasets. These include two public brain imaging datasets, ABIDE and ADHD-200, one public breast cancer imaging dataset, CMMD, and two self-constructed multimodal datasets for breast cancer and gastric cancer. These datasets cover different disease types, modality combinations, and data scales, enabling systematic evaluation of model robustness and scalability in multimodal alignment, fusion, and diagnostic classification. Below, we provide detailed descriptions of the data sources, modality compositions, task settings, and data splits for each dataset.

### D.1. ABIDE

ABIDE is a widely used multi-center brain imaging benchmark dataset for autism spectrum disorder (ASD). It contains neuroimaging data and non-imaging phenotypic information collected from multiple acquisition sites worldwide. The dataset focuses on a binary classification task between ASD subjects and healthy controls (HC), aiming to advance disease identification and mechanism analysis based on multimodal brain imaging. Since the data are collected from different sites, imaging parameters and demographic characteristics vary across subjects. This also makes ABIDE an important benchmark for evaluating the generalization ability of multimodal methods.

To ensure fair comparison with existing state-of-the-art methods, we follow the standard experimental setting and select 871 subjects from the dataset. This includes 468 healthy controls (HC) and 403 ASD patients. The dataset contains rs-fMRI imaging modality as well as non-imaging information, such as demographic features. In this study, the dataset is mainly used to evaluate the stability and alignment capability of the proposed method in complex cross-site brain imaging scenarios.

### D.2. ADHD-200

ADHD-200 is a commonly used multi-center neuroimaging dataset for attention deficit hyperactivity disorder (ADHD) diagnosis studies. It contains resting-state functional MRI (rs-fMRI) data and corresponding non-imaging information. Similar to ABIDE, ADHD-200 exhibits pronounced cross-site heterogeneity. This places higher demands on the robustness and generalization of multimodal models. The task objective of this dataset is to classify healthy controls (HC) and ADHD patients.

Following common experimental protocols, we select data from four sites: New York University Medical Center, Peking University, Kennedy Krieger Institute, and the University of Pittsburgh. Considering that some samples contain missing modalities or incomplete information, we retain 582 subjects after data cleaning, including 364 healthy controls (HC) and 218 ADHD patients.

### D.3. CMMD

CMMD is a large-scale breast X-ray mammography imaging database designed for breast disease research. It contains 3,728 mammographic images from 1,775 patients. The dataset provides rich clinical attribute information, such as age, lesion location, and lesion type. In addition, molecular subtype labels are available for a subset of malignant cases, enabling multimodal prediction tasks for breast cancer molecular subtyping.

In the experiments, we focus on the subset of samples with molecular subtype annotations. Among 749 malignant cases, a total of 1,498 images are associated with molecular subtype information, such as Luminal A, Luminal B, HER2-enriched, and Triple-negative. Since breast cancer molecular subtyping is closely related to prognosis and treatment planning, this task has strong clinical significance. On the CMMD dataset, our study mainly evaluates the effectiveness of the proposed method in real-world imaging diagnosis tasks, especially its ability to jointly model complementary evidence in multimodal settings that combine imaging and clinical information.

## D.4. Self-constructed Breast Cancer Multimodal Dataset

To further evaluate the applicability of the proposed method in real clinical multimodal scenarios, we collaborated with Yuhuangding Hospital to construct a multimodal breast cancer dataset. This dataset contains four modalities from 255 breast cancer patients, including ultrasound images, MRI images, pathological slides, and structured clinical text information. Compared with public datasets, this dataset exhibits more complex modality structures and stronger clinical realism. It is suitable for assessing the model's ability to fuse complementary information across multiple modalities.

Specifically, the ultrasound modality provides one high-quality image per patient, which is used to capture dynamic texture features of breast tissue and tumor regions. The MRI modality contains approximately 200 slices per patient. It characterizes tumor morphology from a three-dimensional perspective and provides spatial relationship information between the tumor and surrounding tissues. The pathology slide modality offers finer-grained lesion evidence at the cellular and histological levels. Each patient provides four high-resolution scanned images at different magnifications. The clinical text modality integrates key medical indicators, including patient age, laboratory test results, tumor size, T stage, and molecular subtype, in a structured format. This dataset is used for breast cancer molecular subtype classification. It also exhibits class imbalance, for example, with 171 Luminal B cases and only 29 triple-negative cases, making it well suited for evaluating robustness and diagnostic performance under imbalanced and complex multimodal conditions. Figure 5 presents several examples.

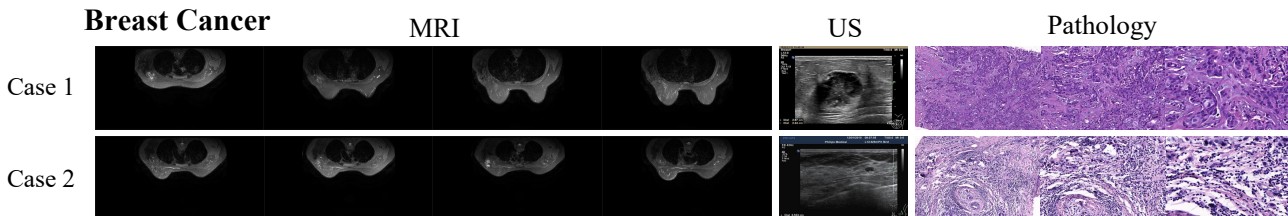

*Figure 5.* Visual examples from the self-constructed breast cancer dataset.

## D.5. Self-constructed Gastric Cancer Multimodal Dataset

To evaluate the generalization ability of the proposed method across different disease types and modality combinations, we further construct a self-constructed gastric cancer multimodal dataset. This dataset contains three modalities from 100 patients, including CT images, gastroscopy images, and clinical text reports. The dataset is designed for gastric cancer classification diagnosis tasks.

In terms of modality composition, the CT modality contains a large number of axial scan slices. To unify input scale and computational cost, we perform slice selection for each patient and retain 200 CT slices. The gastroscopy modality provides 10 images from different viewpoints for each patient, which are used to characterize lesion surface morphology and local visual features. The text modality consists of radiology reports and clinical records that contain diagnosis-related descriptions. Labels for this dataset are derived from pathological diagnosis results and include five classes: adenocarcinoma, poorly differentiated carcinoma, signet-ring cell carcinoma, mucinous carcinoma, and no obvious malignancy. This multi-class task further enables evaluation of the model's ability in multimodal evidence complementarity and fine-grained diagnostic classification. Figure 6 presents several examples.

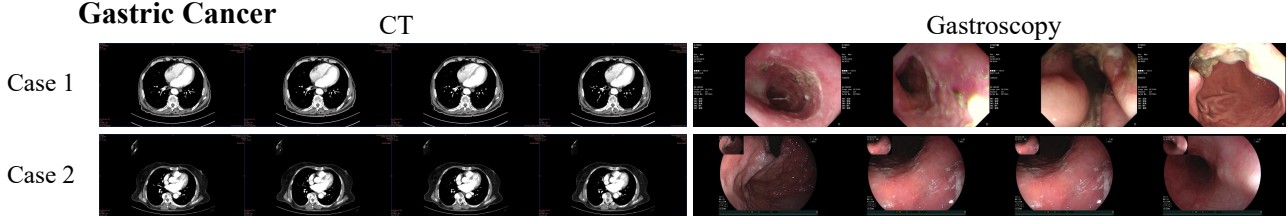

*Figure 6.* Visual examples from the self-constructed gastric cancer dataset.

# E. Parameter Sensitivity Analysis for $\lambda$

In the token-level bidirectional consistency alignment module (Sec. 3.2), we construct cross-modal linear reconstruction operators using a closed-form least-squares solution with L2 regularization. Here, $\lambda$ is the ridge regression regularization coefficient. It is used to alleviate potential ill-conditioning of matrix $H_i^{(k)T} H_i^{(k)}$ and to improve numerical stability during inversion. Since $\lambda$ directly affects the stability and fitting capacity of the reconstruction operator, it further influences the optimization of the token-level consistency constraint. Therefore, it is necessary to conduct a parameter sensitivity analysis on $\lambda$ to select an optimal value.

In our experiments, all other training and evaluation settings are fixed. Only the value of $\lambda$ is varied. Considering the feature scales of medical data, we evaluate a set of continuous strength values: $\lambda \in \{0.1, 0.3, 0.5, 0.7, 0.9\}$.

To intuitively illustrate the overall impact of different values of $\lambda$ on model performance, we visualize the experimental results using radar charts. Specifically, we draw radar charts for four representative datasets. Each radar chart contains four evaluation dimensions: accuracy, sensitivity, specificity, and AUC. In each chart, a closed polygon corresponds to a specific value of $\lambda$. Its covered area and shape reflect the overall performance of the model across the four metrics under that parameter setting. The experimental results are shown in Figure 7.

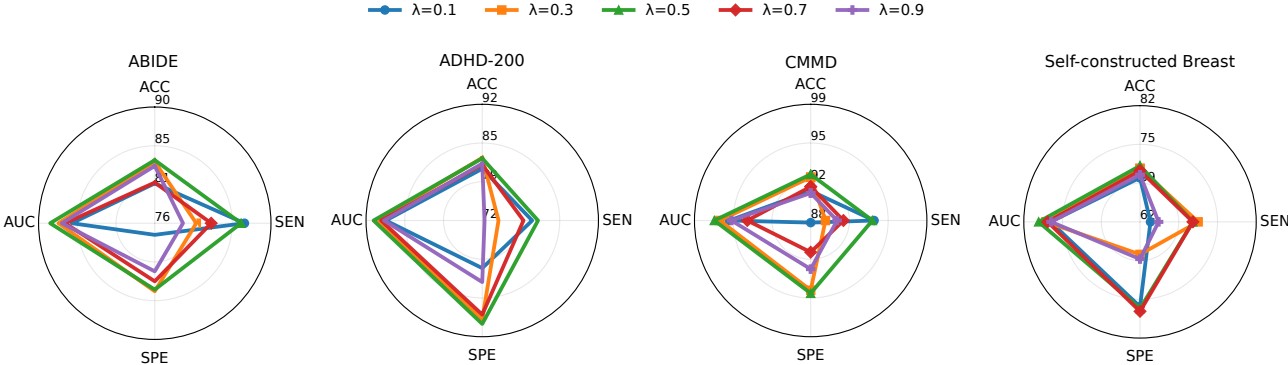

*Figure 7.* Parameter sensitivity analysis for $\lambda$.

From the overall trends of the radar charts, the proposed method exhibits relatively stable performance patterns under the five values of $\lambda$, with limited differences in polygon areas. This indicates that the ridge regularization term in Equation (5) is not a highly sensitive hyperparameter, and the model can maintain reliable alignment and fusion performance over a relatively wide regularization range. When $\lambda$ increases from 0.1 to 0.5, the coverage area of the radar charts expands across datasets and becomes closer to a balanced quadrilateral shape. This suggests that moderate regularization effectively improves the estimation quality of cross-modal linear reconstruction operators. It makes the cross-modal linear interpretability between token representations more stable. As a result, the role of the bidirectional consistency constraint in fine-grained semantic structure alignment is strengthened.

In contrast, when $\lambda$ is further increased to 0.7 or 0.9, the radar chart area slightly shrinks on some datasets. Certain metrics, especially SEN or AUC, are more likely to decline. This trend indicates that overly strong regularization excessively restricts the expressive capacity of the reconstruction operators. As a result, they tend to degenerate toward suboptimal linear mappings, which weakens the ability of the token-level alignment module to model cross-modal local semantic mutual verification. The variation trends of the radar charts are consistent across different datasets. Both public brain imaging datasets and breast cancer datasets exhibit the same pattern, where moderate regularization yields optimal performance, while extreme regularization leads to degradation. This observation further suggests that the role of $\lambda$ mainly lies in controlling numerical stability and the effective rank of the reconstruction operators, rather than being dependent on specific diseases or modality combinations. Overall, the radar charts with $\lambda = 0.3$ generally show the largest area and a more balanced shape.

Based on the observations from the four radar charts, we conclude that the regularization coefficient $\lambda$ in Equation (5) exhibits good robustness within the proposed framework. The model is not sensitive to variations in $\lambda$ and achieves the most stable and balanced performance in a moderate regularization range. Therefore, we adopt $\lambda = 0.3$ as the default setting in all experiments. This sensitivity analysis further validates the feasibility of the proposed token-level bidirectional consistency alignment module in numerical implementation and on real clinical multimodal data. It also supports the reproducibility and

deployment stability of the method.

# F. Quantitative Analysis of Ablation Results

This section provides a quantitative analysis of the results from three core ablation studies to further verify the effectiveness of each module in the proposed method. The ablation studies correspond to three key components of the framework: modality-level semantic uncertainty modeling (Sec. 3.1), token-level bidirectional consistency constraints (Sec. 3.2), and the multi-view consistent fusion with signal–noise suppression mechanism (Sec. 3.3). The corresponding results are summarized in Table 8, Table 9, and Table 10, respectively. Overall, the full model achieves the best comprehensive performance metrics, such as ACC and AUC, across multiple datasets. It also maintains a more stable balance between sensitivity (SEN) and specificity (SPE). These results indicate that the proposed hierarchical consistency alignment and fusion framework not only improves classification performance, but also significantly enhances robustness and generalization in complex clinical scenarios.

## F.1. Effectiveness Analysis of Semantic Uncertainty Modeling

Table 8 compares the ablation results with and without semantic uncertainty modeling. The experiments show a consistent performance improvement after introducing semantic uncertainty across multiple medical multimodal datasets. This improvement is especially reflected in higher ACC and AUC values, as well as a more balanced trade-off between SEN and SPE. Compared with the model w/o Uncertainty, the full model reduces the risks of missed diagnoses and misdiagnoses on multiple datasets. On the one hand, higher sensitivity indicates that the model captures positive cases more effectively. On the other hand, specificity remains more stable, suggesting that incorporating multimodal information does not introduce additional false positives. These results indicate that, during modality-level alignment, treating each modality's semantic representation as a deterministic vector makes the model prone to over-aligning noisy modalities during optimization. This can undermine the stability of the shared semantic space. In contrast, by modeling each modality's estimation of latent semantics as a probabilistic distribution with uncertainty, the proposed method enables adaptive adjustment of alignment strength during training. Reliable modalities contribute more strongly to alignment, while the influence of noisy modalities is naturally attenuated. This significantly improves the stability of cross-modal global semantic alignment and enhances downstream fusion and classification performance.

We further discuss the potential issue of uncertainty collapse in the modality-level symmetric KL alignment. The optimization is based on the full symmetric KL divergence, which contains two coupled terms. Therefore, the loss cannot be continuously reduced simply by arbitrarily scaling the variance of one modality, since severe variance imbalance would significantly enlarge the second term. In implementation, the variance is not treated as an independent free parameter, but is dynamically predicted from the modal input and jointly trained with the corresponding semantic features as an input-dependent reliability estimate. This design also helps alleviate uncertainty collapse. Table 8 further shows that introducing semantic uncertainty modeling consistently improves performance on both the CMMD dataset and the self-constructed breast cancer dataset.

*Table 8.* Ablation study on semantic uncertainty modeling (Sec. 3.1) on CMMD and Self-constructed Breast datasets. (bold: optimal)

| Setting | CMMD | | | | Breast Cancer (Self-constructed) | | | |
| --- | --- | --- | --- | --- | --- | --- | --- | --- |
| | ACC (%) | SEN (%) | SPE (%) | AUC (%) | ACC (%) | SEN (%) | SPE (%) | AUC (%) |
| w/o Uncertainty (Deterministic) | 90.98 (1.01) | 89.12 (1.62) | 92.44 (1.09) | 96.02 (0.78) | 68.92 (1.58) | 66.47 (2.21) | 71.35 (2.04) | 76.88 (1.35) |
| Full (Ours) | **92.37 (0.86)** | **91.84 (1.06)** | **92.89 (0.98)** | **97.12 (0.63)** | **71.64 (1.42)** | **69.10 (2.06)** | **73.92 (1.88)** | **79.45 (1.17)** |

## F.2. Validation of Token-Level Bidirectional Consistency Constraint

Table 9 presents the ablation comparison for the token-level alignment mechanism. Compared with the ablation model that performs cross-modal alignment using local feature cosine similarity, the full model achieves superior classification performance across multiple datasets. The improvements are more consistent on comprehensive metrics such as ACC and AUC. This observation indicates that relying solely on local feature similarity for alignment has clear limitations.

In medical multimodal settings, local token representations from different modalities often exhibit strong structural heterogeneity. For example, local slice structures in MRI do not directly correspond to cellular structures in pathology images. Textual tokens also cannot be simply matched with visual tokens. As a result, similarity-based measures fail to stably capture fine-grained cross-modal semantic correspondences. When substantial structural differences exist between

modalities, cosine-similarity-based alignment becomes more sensitive to local noise, scale variations, and representation shifts. This leads to token-level interactions that lack structural consistency. In contrast, the proposed token-level bidirectional consistency constraint constructs cross-modal explanation operators through linear reconstruction. It further enforces bidirectional closed-loop consistency at the operator level. This design enables local semantic mutual verification from a structural perspective. The mechanism emphasizes that different modalities should not only appear similar. They must also satisfy mutually consistent and interpretable relationships when viewing "A from B" and "B from A". As a result, the reliability and stability of cross-modal local semantic alignment are significantly enhanced. Quantitative results show that introducing this module leads to more robust performance on comprehensive metrics. This demonstrates that fine-grained alignment can be effectively translated into higher-quality fused representations. Consequently, the model achieves stronger discriminative capability.

*Table 9.* Ablation study on token-level bidirectional consistency (Sec. 3.2) on CMMD and Self-constructed Breast datasets. (bold: optimal)

| Setting | CMMD | | | | Breast Cancer (Self-constructed) | | | |
|---|---|---|---|---|---|---|---|---|
| | ACC (%) | SEN (%) | SPE (%) | AUC (%) | ACC (%) | SEN (%) | SPE (%) | AUC (%) |
| w/ Cosine Similarity (Replace Sec. 3.2) | 91.18 (0.96) | 90.06 (1.33) | 92.01 (1.05) | 96.21 (0.71) | 69.74 (1.51) | 66.92 (2.18) | 72.33 (1.91) | 77.62 (1.22) |
| Full (Ours) | **92.37 (0.86)** | **91.84 (1.06)** | **92.89 (0.98)** | **97.12 (0.63)** | **71.64 (1.42)** | **69.10 (2.06)** | **73.92 (1.88)** | **79.45 (1.17)** |

## F.3. Validation of Multi-View Consistent Fusion and Signal–Noise Suppression

Table 10 presents a systematic ablation analysis of the fusion module in Sec. 3.3. The results show that the full model achieves the best performance across all metrics. Compared with w/o Multi-view, removing the multi-view mechanism leads to a clear degradation in overall performance. This demonstrates that multi-view conditional fusion is not a redundant fusion scheme. Instead, it is able to extract more consistent shared semantics and complementary evidence from different conditional modality views. The key advantage of the multi-view mechanism is that the fusion process does not rely on a fixed primary modality. By alternately selecting each modality as the conditional view, every modality can act as a semantic reference at a certain fusion stage. This design promotes the formation of a cross-modal consensus representation in the final fused result. Compared with w/o Denoise, removing signal–noise decomposition and noise suppression leads to a significant decrease in SPE. This indicates that noise components introduce more false positives and weaken classification reliability. This conclusion is consistent with practical risks in medical diagnosis. Noisy information often causes the model to misclassify healthy samples as positive, thereby reducing specificity and increasing the cost of clinical misdiagnosis. Finally, w/o Both, which removes both the multi-view mechanism and denoising, results in the most pronounced performance degradation. This shows that the two mechanisms are not interchangeable, but complementary components in the fusion stage. The multi-view mechanism improves the integration efficiency of complementary multimodal information. It allows the fused representation to cover more comprehensive diagnostic evidence. The denoising mechanism further suppresses redundancy and noise-induced negative transfer. This ensures that the fusion process is not disturbed by low-quality information. Together, these two mechanisms enable the model to exhibit stronger stability and generalization on complex breast cancer multimodal data with multiple sources and significant modality quality variations.

*Table 10.* Ablation study on multi-view consistency fusion and signal–noise suppression (Sec. 3.3) on CMMD and Self-constructed Breast datasets. (bold: optimal)

| Setting | CMMD | | | | Breast Cancer (Self-constructed) | | | |
|---|---|---|---|---|---|---|---|---|
| | ACC (%) | SEN (%) | SPE (%) | AUC (%) | ACC (%) | SEN (%) | SPE (%) | AUC (%) |
| w/o Multi-view | 90.94 (1.02) | 90.71 (1.24) | 91.22 (1.07) | 95.98 (0.80) | 68.87 (1.63) | 67.55 (2.09) | 70.46 (2.14) | 76.92 (1.26) |
| w/o Denoise | 91.22 (0.98) | **92.03 (1.11)** | 89.76 (1.26) | 96.18 (0.73) | 69.32 (1.57) | **70.14 (1.98)** | 68.21 (2.35) | 77.28 (1.21) |
| w/o Both | 89.76 (1.10) | 90.05 (1.36) | 89.41 (1.30) | 95.20 (0.88) | 67.41 (1.71) | 66.38 (2.17) | 68.72 (2.28) | 75.83 (1.34) |
| Full (Ours) | **92.37 (0.86)** | 91.84 (1.06) | **92.89 (0.98)** | **97.12 (0.63)** | **71.64 (1.42)** | 69.10 (2.06) | **73.92 (1.88)** | **79.45 (1.17)** |

In summary, the three ablation studies quantitatively validate the hierarchical consistency design of the proposed framework. The uncertainty modeling in Sec. 3.1 provides a reliability-adaptive mechanism for modality-level alignment. The token-level bidirectional consistency in Sec. 3.2 further strengthens fine-grained local semantic mutual verification. The multi-view fusion and denoising mechanism in Sec. 3.3 efficiently extracts complementary signals and suppresses noise within a unified

semantic space. Together, these components form a closed-loop alignment and fusion pipeline, offering a robust, scalable, and theoretically motivated solution for multimodal medical diagnosis tasks.

## G. Computational Cost Analysis

We further analyze the computational cost of the proposed method with respect to the number of modalities. The additional computation mainly comes from two parts. First, the token-level bidirectional consistency constraint requires computation across modality pairs. Second, the multi-view fusion strategy requires each modality to be used in turn as the conditional view for shared semantic extraction and fusion. However, this additional overhead mainly arises in the consistency constraint and fusion stages, rather than in the backbone encoding network itself. Meanwhile, the token-level reconstruction adopts a compact linear formulation and does not introduce additional complex nonlinear mapping modules. Therefore, the computational cost for each modality pair remains relatively manageable. To quantitatively verify this, we report the corresponding changes in FLOPs as the number of modalities increases in Table 11.

*Table 11.* FLOPs of the proposed method under different numbers of modalities.

| Number of Modalities | FLOPs (GFLOPs) |
| --- | --- |
| 2 | 0.0105 |
| 3 | 0.0151 |
| 4 | 0.0198 |
| 5 | 0.0245 |

As shown in Table 11, although the computational cost increases as the number of modalities grows, the overall growth remains moderate. Even when five modalities are used, the FLOPs are only 0.0245 GFLOPs. This indicates that the proposed method introduces limited additional computational overhead in multimodal settings. In practical medical multimodal tasks, the number of available modalities is usually small, so the additional cost caused by pairwise consistency modeling and multi-view fusion remains manageable and does not impose a significant computational burden.

