# OpenReview forum: "Scalable Medical Multimodal Fusion via Symmetric Consistency Modeling"
_ICML.cc/2026/Conference — ICML 2026 regular_

### Official Review · Reviewer_cVrW · 2026-03-02

**Soundness:** 3
**Presentation:** 3
**Significance:** 2
**Originality:** 2
**Overall Recommendation:** 4
**Confidence:** 2

**Summary:**

Medical multi-modal data contain semantic homogeneity and different reliability. To tackle this problem, the authors propose SMMF, which is a scalable multi-modal fusion framework. It has three main components. Modality-level consistency alignment maintains the distribution consistency of different modalities. Token-level bidirectional consistency alignment to ensure local information. Finally, the multi-view consistent fusion models fused features to signals and noise.

**Compliance With Llm Reviewing Policy:**

Affirmed.

**Key Questions For Authors:**

Please see the weaknesses mentioned above. My most concern is about the uavailable URL.

**Limitations:**

yes

**Strengths And Weaknesses:**

Strengths:

S1- The paper is well structured and makes a clear presentation.

S2- The authors design consistency enhancement mechanisms from different aspects, including the modality level, the token level, and the multi-view fusion.

S3- Sufficient experiments are conducted to verify the effectiveness of the designs. Experiments on both public and self-collected datasets show the power of SMMF.

Weaknesses:

W1- The URL of the code is not available. This undermines the reproducibility of the paper.

W2- Some typos should be fixed. For example, in Table 1, the second-best score of SPE of ADHD-200 is not underlined.

W3- According to Eq.10, the final fusion is accomplished by simply adding up. How about other fusion methods, such as MLP or a cross-attention?

W4- If the authors can compare other baselines with different modality numbers in Table 4, this experiment will verify the effectiveness of the symmetric consistency modeling better.

---

> ### Author Rebuttal · Authors · 2026-03-31
>
> # Response to Reviewer #4
>
> We sincerely appreciate the constructive suggestions from reviewers. We will explain your concerns point by point.
>
> **Q1: The URL of the code is not available. This undermines the reproducibility of the paper.**
>
> A1: We sincerely apologize, and your comment has just made us aware of this issue. Because we previously requested a username change for our GitHub account, the account is currently in a locked state. Our testing shows that the page is accessible only when we are logged into the account ourselves. We have already reported this issue to the GitHub community. We are very sorry for the inconvenience. To resolve this, we have uploaded the code to another GitHub account at: https://github.com/xxiaowSun/SMMF.
>
> **Q2: Some typos should be fixed. For example, in Table 1, the second-best score of SPE of ADHD-200 is not underlined.**
>
> A2: We rechecked the entire manuscript and corrected the related typographical and formatting errors. Specifically, the incorrect boldface/underline markings were fixed in Tables 1, 2, 3, and 7. These revisions do not affect the reported results or conclusions.
>
> **Q3: According to Eq.10, the final fusion is accomplished by simply adding up. How about other fusion methods, such as MLP or a cross-attention?**
>
> A3: The focus of this work is not on using a complex fusion module in the final step. Instead, before the final aggregation, we refine shared high-value information across modalities through multi-view consistency modeling, signal/noise decomposition, and conditional view constraints. Therefore, the summation in Eq. (10) is not a direct aggregation of raw heterogeneous features, but a combination of shared representations after consistency optimization. Under this premise, we adopt simple aggregation to avoid additional complex parameters that may affect the effectiveness of the preceding multi-view consistency modeling, while maintaining stability and interpretability in the fusion stage.
>
> To support this claim, we include an additional ablation study in the revised manuscript, comparing three final aggregation strategies: (1) MLP-based fusion, (2) cross-attention-based fusion, and (3) Ours.
>
> | Method | CMMD ACC / AUC | In-house Breast Cancer ACC / AUC |
> |---|---|---|
> | MLP-based fusion | 91.86 / 96.68 | 70.52 / 78.62 |
> | Cross-attention-based fusion | 92.01 / 96.84 | 70.88 / 78.95 |
> | Ours | 92.37 / 97.12 | 71.64 / 79.45 |
>
> The results show that MLP and cross-attention model inter-modal interactions by introducing additional parameters, but their overall classification performance is slightly lower than that of the proposed method. This further demonstrates that multi-view consistency modeling has already effectively extracted high-quality shared signals. The current fusion strategy can more stably integrate useful information without requiring additional complex parameterized aggregators.
>
> **Q4:  If the authors can compare other baselines with different modality numbers in Table 4, this experiment will verify the effectiveness of the symmetric consistency modeling better.**
>
> A4: We thank the reviewer for this constructive suggestion. Reporting only the performance of the proposed method under different numbers of modalities is not sufficient to fully demonstrate the advantages of symmetric consistency modeling in multimodal scalability scenarios. Considering that Meta-Transformer achieves strong performance under its original settings on the in-house breast cancer and gastric cancer datasets, we extend it to a multimodal version that can handle variable modality inputs. We then compare it with the proposed method under the same modality combination settings as in Table 4. In implementation, we keep the core architecture of Meta-Transformer unchanged and only modify its input and fusion components to accept a variable number of modality tokens. We then train and evaluate it under different modality combinations.
>
> The experimental results show that as the number of modalities increases, both the adapted Meta-Transformer and the proposed method benefit from additional modality information. However, the proposed method consistently achieves better performance under different modality settings and exhibits a more stable improvement trend as the number of modalities increases. Compared with Meta-Transformer, the proposed method achieves consistent improvements under different modality settings.
>
> | Dataset | Modality Setting | Meta-Transformer (ACC / AUC) | Ours (ACC / AUC) |
> |---|---|---|---|
> | Breast Cancer | {1,2} | 70.42 / 78.36 | 71.64 / 79.45 |
> | Breast Cancer | {1,2,3} | 72.61 / 80.76 | 73.92 / 82.04 |
> | Breast Cancer | {1,2,3,4} | 74.03 / 82.61 | 75.38 / 83.97 |
> | Gastric Cancer | {1,2} | 71.98 / 80.17 | 73.11 / 81.26 |
> | Gastric Cancer | {1,2,3} | 73.56 / 81.94 | 74.62 / 82.73 |
>
> We have added these results in the revised manuscript to further validate the effectiveness of the proposed method in multimodal scalability scenarios.

---

> > ### Author Rebuttal · Reviewer_cVrW · 2026-04-01
> >
> > The authors have addressed all my questions. Considering the significance and originality of the paper, I'm going to keep my score.

---

> > > ### Author Response · Authors · 2026-04-07
> > >
> > > Thank you for taking time out of your busy schedule to help us improve the quality of our paper. We will carefully revise and rigorously check the issues you raised to ensure further improvement in the quality of the paper. Thank you again.

---

### Official Review · Reviewer_5Rpn · 2026-03-05

**Soundness:** 3
**Presentation:** 3
**Significance:** 2
**Originality:** 3
**Overall Recommendation:** 4
**Confidence:** 3

**Summary:**

This paper proposes a modality-agnostic medical multimodal fusion framework. Its core contributions lie in three aspects:
1. Introducing the symmetric KL divergence with uncertainty perception at the modality level for global semantic alignment.
2. Employing deterministic linear reconstruction (closed-form solution) at the token level to enforce bidirectional consistency constraints.
3. Employing a multi-view strategy based on conditional mutual information and conditional entropy during the fusion stage to suppress noise.

**Compliance With Llm Reviewing Policy:**

Affirmed.

**Final Justification:**

The author's rebuttal addressed most of my concerns, and I will maintain my score.

**Key Questions For Authors:**

1. Section 3.1 models semantic features as a diagonal Gaussian distribution with learnable parameters $\sigma$. When minimizing the symmetric KL divergence (Equation 14), without supervision from true variance labels, the network is prone to easily minimize the precision weighting term by predicting all $\sigma$ to be infinite ($\sigma \to \infty$), thereby falling into the trivial solution of uncertainty collapse. How can this issue be prevented?
2. Why provide a code link without providing the code?
3. Other questions have already been addressed under weaknesses.

**Limitations:**

yes

**Strengths And Weaknesses:**

**Strengths**
1. The paper recognizes that single-level alignment is insufficient for handling complex medical data, thus it is reasonable to design a multi-level consistency framework progressing from modality level to token level and then to the fusion layer.
2. The authors not only utilized publicly available datasets such as ABIDE, ADHD-200, and CMMD, but also introduced two self-built multimodal datasets (breast cancer and gastric cancer) for validation. Extensive ablation experiments provided support for the effectiveness of each module.
3. The writing of the paper is good.
4. It is meaningful to combine the semantic alignment of uncertain perceptions with deterministic token-level linear reconstruction, rather than relying solely on attention mechanisms or cosine similarity for local alignment.

**Weaknesses**
1. Equation (8) introduces conditional mutual information and conditional entropy as optimization objectives. However, the precise computation of mutual information for high-dimensional continuous features is highly challenging. The paper does not elaborate on how these metrics are estimated in the code implementation.
2. Token-level bidirectional alignment employs a closed-form reconstruction operator (Equation 5). However, if the token sequence length $L_k$ is smaller than the feature dimension $d_t$, the covariance matrix inevitably becomes rank-deficient. In this case, the inversion process is entirely dominated by the regularization term $\lambda$. The paper fails to provide specific details regarding the dimensions of $L_k$ and $d_t$, nor does it mathematically demonstrate the absolute validity of this closed-form solution in such feature spaces.
3. This is a paper claiming the method is effective, yet it lacks any pseudocode or key hyperparameter settings (such as feature dimension, specific optimizer implementation details) for implementing the aforementioned core complex mathematical formula.
4. I spent a lot of time analyzing Figure 1, but it's difficult to understand the author's purpose based only on Figure 1.

---

> ### Author Rebuttal · Authors · 2026-03-31
>
> # Response to Reviewer #3
>
> We sincerely appreciate the constructive suggestions from reviewers. We will explain your concerns point by point.
>
> **Q1: Equation (8) uses conditional mutual information and conditional entropy as objectives, but the paper does not clearly explain how they are estimated in practice for high-dimensional continuous features.**
>
> A1: We realized that the original manuscript did not sufficiently explain the practical implementation of Equation (8). In practice, we first decompose each modal representation into a signal part to be preserved and a noise part to be suppressed. The mutual information objective is approximated by an InfoNCE-based contrastive loss, which encourages the extracted signal from other modalities to align with the current conditional view. For the conditional entropy term, we implement it indirectly by imposing constraints on the noise component. If a part of the representation corresponds to noise or redundant information, it should not remain highly correlated with the current conditional view during modeling. Based on this principle, we approximate the conditional entropy objective by compressing the noise component.
>
> **Q2: The paper does not specify $L_k$ and $d_t$, nor justify the closed-form reconstruction when $L_k<d_t$ makes the matrix rank-deficient.**
>
> A2: In practice, we avoid constructing the reconstruction operator directly in the original high-dimensional feature space by using an MLP-based tokenizer, which maps both the token sequence length and the token feature dimension of each modality to 8. Under this setting, Eq. (5) is solved in a low-dimensional token subspace with matched dimensions. More importantly, this closed-form solution is not intended to recover an exact inverse mapping, but to construct a regularized and stable linear reconstruction operator for characterizing cross-modal local structural relations. With the $\lambda$ term, Eq. (5) becomes a ridge-regularized least-squares problem, where $H_i^{(k)T}H_i^{(k)}+\lambda I$ is always positive definite, ensuring the existence and uniqueness of the solution. Appendix E further shows that the model remains stable over a range of $\lambda$ values. We have summarized this discussion in the revised manuscript as part of the analysis of Eq. (5).
>
> **Q3: The paper claims strong effectiveness, but lacks pseudocode and key hyperparameter details needed to implement its core formulas.**
>
> A3: We have added more implementation details in Sec. 4.1 of the revised manuscript and included pseudocode in the appendix. Specifically, we now clarify that the modality representation dimension $d$ in Sec. 3.1 is set to 128, while the token sequence lengths $L_k$, $L_m$, and the token feature dimension $d_t$ in Sec. 3.2 are all set to 8. We also supplement the optimizer and training settings, explicitly stating that Adam is used. In addition, we provide pseudocode for the overall method, covering the main steps of modality encoding, modality-level uncertainty modeling, tokenization and bidirectional reconstruction, multi-view signal/noise decomposition, and end-to-end optimization with the downstream classification objective.
>
> **Q4: I spent a lot of time analyzing Figure 1, but it's difficult to understand the author's purpose based only on Figure 1.**
>
> A4: To present the content of Figure 1 and our research objective more clearly, we rewrote the figure caption. The revised caption first explicitly states that the figure is intended to illustrate how robust multimodal fusion is achieved by progressively establishing cross-modal consistency. It then further clarifies the core role of the upper, middle, and lower parts, which correspond to global semantic distribution alignment, local structural consistency modeling, and multimodal fusion for downstream medical classification, respectively.
>
> **Q5: Section 3.1 may suffer from uncertainty collapse, as minimizing the symmetric KL without variance supervision could drive all σ toward infinity.**
>
> A5: Our optimization is based on the full symmetric KL divergence, which contains two coupled terms. Therefore, the loss cannot be continuously reduced simply by arbitrarily scaling the variance of one modality, since severe variance imbalance would significantly enlarge the second term. In implementation, the variance is not treated as an independent free parameter, but is dynamically predicted from the modal input and jointly trained with the corresponding semantic features as an input-dependent reliability estimate. This design also helps alleviate uncertainty collapse. The ablation results in Appendix F.1 further support this point, showing consistent improvements on both the CMMD dataset and the in-house breast cancer dataset after introducing semantic uncertainty modeling. We have added this discussion to the first part of the ablation study in the revised manuscript.
>
> **Q6: Why provide a code link without providing the code?**
>
> A6: Please refer to A1 to reviewer#4.

---

> > ### Author Rebuttal · Reviewer_5Rpn · 2026-04-01
> >
> > I appreciate the additional clarifications in the rebuttal. I think Q2, Q4 and Q5 are reasonably addressed, but Q1and Q3 remain insufficiently resolved.
> >
> > * For Q1, the rebuttal explains that Eq. (8) is implemented via an InfoNCE-style surrogate plus noise compression, but it still does not clearly specify the conditioning, negative construction, or why this surrogate faithfully matches the stated objective.
> > * For Q3, the added implementation details are helpful, but reproducibility concerns remain because the released code still appears inconsistent with the manuscript in key settings and also includes several extra training heuristics that could affect the reported results.
> > * Regarding the code, have you checked the code you submitted?
> >
> > The rebuttal improves clarity, but it does not fully resolve the methodological and reproducibility concerns.

---

> > > ### Author Response · Authors · 2026-04-05
> > >
> > > **Q1: A more detailed explanation of the code implementation process and related principle verification is needed for Eq. (8).**
> > >
> > > A1: In order to better illustrate the ideological logic of the proposed method, we have provided a more detailed introduction for Eq. (8).
> > >
> > > (1) For mutual information, the effective information extracted from other modalities is denoted as $S$, and the current conditional view is denoted as $C$. At this point, the process of utilizing mutual information to preserve high-value shared information in Eq. (8) can be expressed as: I(S;C) =H(S)-H(S|C).
> > >
> > > It can be seen that if $S$ and $C$ have a large amount of high-value shared information, then given $C$, the uncertainty of $S$ should be reduced. InfoNCE improves the ability of conditional views to identify high-value shared information by narrowing positive sample pairs and separating negative sample pairs. The implementation code is as follows:
> > >
> > > logits = q @ k.t() / self.temp
> > >
> > > labels = torch.arange(q.size(0), device=q.device)
> > >
> > > return F.cross\_entropy(logits, labels)
> > >
> > > During training, within the same batch, $(q\_i, k\_i)$ from the same sample is treated as a positive pair, while $(q\_i, k\_i),i\\ne j$ from different samples are treated as negative pairs. Therefore, InfoNCE encourages the model to bring the current conditional view closer to the informative content contained in its corresponding positive pair. In this sense, it approximates the objective of maximizing high-quality shared information and can therefore serve as a surrogate for mutual information.
> > >
> > > (2) For conditional entropy, this paper indirectly achieves it through noise suppression. Denoting the noise component as $N$, conditional entropy H(S|Z) represents the residual uncertainty of $S$ given $Z$. This implies that if we wish to reduce the uncertainty of $S$, on one hand, we need to compress the amplitude of $N$ itself. And on the other hand, we need to remove as much information from $Z$ that is unrelated to the conditional view, ensuring that the remaining information $S$ is highly correlated with the conditional view. Based on this principle, our implementation adopts two types of constraints simultaneously: noise compression and noise decorrelation:
> > >
> > > L_noise = (n_img.pow(2).mean() + n_txt.pow(2).mean()) + (torch.abs((n_img_n * txt_n).sum(dim=-1)).mean() + torch.abs((n_txt_n * img_n).sum(dim=-1)).mean())
> > >
> > > **Q2: More detailed explanations are needed for code inspection and reproducibility.**
> > >
> > > A2: We have rechecked which important details in the code were omitted in the paper. In the code we provided, the model file contains a total of 9 classes and 3 functions.
> > >
> > > GaussianUncertaintyHead and symmetric\_kl\_diag\_gaussian implement Eqs. (1) and (2). Tokenizer, ridge\_reconstruction\_operator, and token\_bidir\_operator\_loss implement Eqs. (4)–(6). SignalNoiseDecomposer and TwoViewFusion correspond to Eqs. (7)–(9). MM\_GTUNets integrates all modules and constructs the overall loss.
> > >
> > > We found that four components, VAE, CustomConv2d, RP\_Attention, and Multimodal\_Attention, were not described in the original manuscript. These modules support the training of the modal encoders. We adopt Graph U-Nets (TPAMI 2021) as the modal encoder because, unlike most multimodal methods that encode each modality independently, our goal is to introduce cross modal interaction already at the encoding stage so as to obtain more discriminative representations.
> > >
> > > The VAE is used to preprocess the text modality, meaning that the input text is first encoded by the VAE and then fed into Graph U-Nets for feature extraction. Compared with directly using the raw text features, the VAE helps preserve the main semantic information while suppressing redundant noise and local perturbations. Both the VAE and Graph U-Nets are retrained in our implementation, and the final loss function is formulated as:
> > >
> > > aux_loss = w_graph * L_graph + w_mod * L_mod + w_tok * L_tok + w_fus * L_fus
> > >
> > > L_mod, L_tok, and L_fus correspond to the losses in 3.1, 3.2, and 3.3, respectively. L_graph corresponds to the loss of Graph U-Nets. w_graph, w_mod, w_tok, and w_fus correspond to the weights of the four loss functions, respectively. We obtained the weights of 1.0, 0.2, 0.01, and 0.2 through grid search.
> > >
> > > CustomConv2d, RP_Attention, and Multimodal_Attention are mainly used to implement the residual connection, which further combines the encoder outputs with the multi-view fusion result in Sec. 3.3. This design is optional and contributes only a minor performance gain, with an accuracy improvement of about 0.2%. Therefore, we kept this implementation in the released code.
> > >
> > > We have organized the above content and included it in the model details section of the revised manuscript. Meanwhile, we have further supplemented the training details directly related to reproducibility. These include specific settings for dropout, the learning rate of the main model, the learning rate during the VAE pre-training stage.

---

### Official Review · Reviewer_G3ev · 2026-03-10

**Soundness:** 3
**Presentation:** 3
**Significance:** 2
**Originality:** 3
**Overall Recommendation:** 3
**Confidence:** 3

**Summary:**

This paper proposes a modality-agnostic medical multimodal fusion framework aimed at overcoming the limitation of traditional methods that require a fixed number of modalities. The framework combines coarse-grained modality-level alignment with fine-grained token-level consistency constraints and employs a multi-view conditional fusion strategy to obtain a unified representation, thereby improving performance in multimodal medical diagnosis tasks. Experimental results show that the proposed method achieves strong performance on multiple public and in-house medical datasets and demonstrates good scalability when handling different modality combinations and varying numbers of modalities.

**Compliance With Llm Reviewing Policy:**

Affirmed.

**Final Justification:**

I appreciate the additional clarifications provided in the rebuttal. While Q2 and Q3 have been addressed reasonably well, Q1 remains insufficiently resolved.

Specifically, the response does not address whether imbalanced modality availability (e.g., different sample numbers or missing rates) may lead to one modality dominating the fusion process in practice. Even if modalities alternate as the conditional view, a modality that is more frequent or more complete may still have a disproportionate influence during optimization. This issue should be clarified further, ideally with additional analysis or experimental evidence.

Overall, the rebuttal improves clarity, but it does not fully resolve this concern. Therefore, I will maintain my original score.

**Key Questions For Authors:**

1.The token-level reversible consistency in the method relies on a linear reconstruction assumption between modalities. Given that cross-modal relationships are inherently highly nonlinear, does this assumption still hold in practice?
2.The method involves pairwise modality alignment and bidirectional token reconstruction, which may cause the computational complexity to grow quadratically with the number of modalities. Could the authors provide a clearer analysis of the computational cost of the proposed model?

**Limitations:**

Yes

**Strengths And Weaknesses:**

Strengths:
1.The paper addresses the limitation of fixed modality numbers in traditional medical multimodal methods. By introducing a multi-view conditional fusion strategy, the proposed framework can naturally extend to an arbitrary number of modalities, showing strong application potential.
2.The method enhances the stability of both local and global cross-modal semantic alignment through symmetric KL consistency constraints and bidirectional reconstruction, which helps mitigate modality inconsistency.
3.The experiments cover multiple public and in-house medical datasets, which validates the effectiveness of the proposed method across diverse medical data types.

Weaknesses:
1.Although the framework supports multiple modalities, the multi-view fusion strategy requires each modality to be alternately used as a conditional view during optimization. This may cause the model to rely heavily on certain modalities, potentially affecting the quality of multimodal fusion.
2.The model design is relatively complex. Although it supports multiple modalities, it requires pairwise modeling between modalities, and the paper does not provide a detailed analysis of computational complexity.

---

> ### Author Rebuttal · Authors · 2026-03-31
>
> # Response to Reviewer #2
>
> We sincerely appreciate the constructive suggestions from reviewers. We will explain your concerns point by point.
>
> **Q1: The multi-view fusion strategy may over-rely on certain modalities, potentially harming fusion quality.**
>
> A1: Regarding the risk of conditional view bias introduced by the multi-view fusion strategy, our design does not allow any single modality to act as a dominant view over time. Instead, each modality serves as a conditional view while simultaneously imposing reciprocal constraints on the others. In our method, each modality takes turns serving as the conditional view during training. This design enables the model to perform multi-directional modeling of shared information from different modalities, rather than aggregating information around a single fixed dominant modality.
>
> In addition, to address this issue, our method further adopts a strategy that separates high-value information from noise, thereby reinforcing the stable shared components across modalities. The results in Table 4 show that as the number of modalities increases, this strategy consistently improves fusion performance, indicating that the model does not degenerate into relying on a single modality. Based on your comment and the above analysis, we have supplemented the experimental analysis of Table 4 in the revised manuscript.
>
> **Q2: The token-level reversible consistency in the method relies on a linear reconstruction assumption between modalities. Given that cross-modal relationships are inherently highly nonlinear, does this assumption still hold in practice?**
>
> A2: In our method, the linear reconstruction constraint is mainly used to characterize local structural consistency within shared semantics. After semantic alignment, the token representations of different modalities have already become comparable to a certain extent. In this context, the role of bidirectional linear reconstruction is to examine whether local semantic structures can mutually explain each other, thereby providing an interpretable constraint signal for token-level consistency. More specifically, our method relies on a local linear approximation, where the aligned semantic structures already provide a basis for similarity analysis. At the same time, this design does not introduce additional trainable parameters, which helps reduce computational overhead. To further validate this point, we replaced the linear reconstruction with an MLP-based nonlinear mapping for comparison.
>
> | Method | CMMD ACC| CMMD SEN | CMMD SPE | CMMD AUC | Breast Cancer ACC | Breast Cancer SEN | Breast Cancer SPE| Breast Cancer AUC |
> |---|---:|---:|---:|---:|---:|---:|---:|---:|
> | MLP | 91.68 | 91.57 | 92.64 | 96.54 | 70.93 | 68.62 | 73.36 | 79.26 |
> | Ours | 92.37 | 91.84 | 92.89 | 97.12 | 71.64 | 69.10 | 73.92 | 79.45 |
>
> The experimental results show that the MLP-based variant performs almost on par with our method, and in some cases even leads to lower classification accuracy. These results further support that our method effectively captures local linear approximation rather than relying on a global linear assumption. We have added a corresponding discussion in the ablation study section of the revised manuscript.
>
> **Q3: The method involves pairwise modality alignment and bidirectional token reconstruction, which may cause the computational complexity to grow quadratically with the number of modalities. Could the authors provide a clearer analysis of the computational cost?**
>
> A3: For our method, the additional overhead mainly comes from two parts. First, the token-level bidirectional consistency constraint requires computation across modality pairs. Second, the multi-view fusion strategy requires each modality to be used in turn as the conditional view for shared semantic extraction and fusion. However, this additional overhead mainly arises in the consistency constraint and fusion stages, rather than in the backbone encoding network itself. Meanwhile, the token-level reconstruction adopts a compact linear formulation and does not introduce additional complex nonlinear mapping modules. Therefore, the computational cost for each modality pair remains relatively manageable.
>
> To verify this point, we further supplement the corresponding changes in FLOPs as the number of modalities increases in the revised manuscript.
>
> | Number of Modalities | FLOPs (GFLOPs) |
> |---|---:|
> | 2 | 0.0105 |
> | 3 | 0.0151 |
> | 4 | 0.0198 |
> | 5 | 0.0245 |
>
> It can be observed that although the computational cost increases with the number of modalities, the overall growth trend remains relatively gradual. Even with five modalities, the FLOPs are only 0.0245 GFLOPs, indicating that the proposed method maintains a low computational cost in multimodal settings. In practical settings, medical multimodal tasks usually involve only a small number of modalities, so this additional overhead remains manageable and does not introduce a significant computational burden.

---

> > ### Author Rebuttal · Reviewer_G3ev · 2026-04-02
> >
> > I appreciate the additional clarifications provided in the rebuttal. While Q2 and Q3 have been addressed reasonably well, Q1 remains insufficiently resolved.
> >
> > Specifically, the response does not address whether imbalanced modality availability (e.g., different sample numbers or missing rates) may lead to one modality dominating the fusion process in practice. Even if modalities alternate as the conditional view, a modality that is more frequent or more complete may still have a disproportionate influence during optimization. This issue should be clarified further, ideally with additional analysis or experimental evidence.
> >
> > Overall, the rebuttal improves clarity, but it does not fully resolve this concern.

---

> > > ### Author Response · Authors · 2026-04-07
> > >
> > > Thank you to the reviewer for raising this crucial question. Our previous response focused on explaining from the perspective of the design mechanism of multi-view fusion. There is a lack of direct experimental data to verify the behavioral characteristics of the model in extreme imbalance scenarios. To address this, we have designed a set of experimental verifications specifically for scenarios with imbalanced modal integrity.
> > >
> > > Firstly, we conducted six sets of ablation experiments using the three-modal setting in our self-constructed breast cancer dataset as a benchmark. The first three sets involved classification using only one modality, establishing the model classification ability when the model completely degenerates to a single modality. To apply the method proposed in this paper, when using a single modality, we replicated it into two modality inputs and randomly occluded 5% of the area in each modality to simulate multi-modal data.
> > >
> > > For the last three sets of experiments, we artificially constructed explicit unbalanced modal scenarios. In the third-modal setting, we kept one of the modalities intact at all times and randomly discarded 30% of the input samples from the other two modalities. The purpose of this was to deliberately create an input situation where one modality is significantly more complete while the other modalities are missing. By comparing the results of these three sets of experiments with those of single-modality experiments, we can verify whether the model is dominated by the complete modality. The experimental results are shown in the table below.
> > >
> > > | Setting                          | ACC | AUC |
> > > |----------------------------------|-----|-----|
> > > | Only Modality A                 | 68.93 | 77.86 |
> > > | Only Modality B                 | 70.19 | 79.08 |
> > > | Only Modality C                 | 69.45 | 78.37 |
> > > | A full, B/C 30% missing         | 71.98 | 80.66 |
> > > | B full, A/C 30% missing         | 73.05 | 81.53 |
> > > | C full, A/B 30% missing         | 72.42 | 81.14 |
> > >
> > > It can be observed that in all scenarios, the model performance not only did not degrade to the level of single modality, but also maintained a significant multimodal fusion gain. This indicates that even with a 30% missing rate in the other two modalities, the model is still able to extract effective complementary information from these partially incomplete modalities. It's not simply relying on the always intact modality to make decisions.
> > >
> > > In the ablation study of the revised manuscript, we have included this experiment and analysis to more explicitly demonstrate the behavioral characteristics and fusion stability of the proposed method under conditions of imbalanced modality availability.

---

### Official Review · Reviewer_jux7 · 2026-03-10

**Soundness:** 2
**Presentation:** 3
**Significance:** 2
**Originality:** 2
**Overall Recommendation:** 4
**Confidence:** 3

**Summary:**

This paper proposes a modality-agnostic multimodal fusion framework for medical diagnosis. The method combines uncertainty-aware modality-level alignment, token-level bidirectional consistency via linear reconstruction, and a multi-view fusion strategy intended to suppress noise and extract shared semantics. Experiments on five medical multimodal datasets, including public and self-constructed benchmarks, show improved classification performance and some evidence of scalability to varying modality numbers.

**Compliance With Llm Reviewing Policy:**

Affirmed.

**Final Justification:**

The additional results address most of my concerns and strengthen the paper, so I will raise my score to weak accept.

**Key Questions For Authors:**

1. Can the authors evaluate the method under explicit missing-modality or unseen modality-subset settings, since this is central to the claimed contribution?
2. Can the authors clarify more precisely what is fundamentally novel relative to prior uncertainty-aware or consistency-based multimodal fusion methods?
3. Can the authors provide calibration and computational cost analysis, especially as the number of modalities increases?

**Limitations:**

Yes. The paper discusses bias, calibration, and deployment concerns.

**Strengths And Weaknesses:**

**Strengths**
1. The paper addresses an important practical problem: multimodal medical data often come with varying modality availability, and the goal of modality-agnostic fusion is well motivated.
2. The method is reasonably coherent, with a clear three-part design: uncertainty-aware global alignment, token-level structural consistency, and multi-view fusion. The overall framework is easy to follow.
3. The empirical section is fairly comprehensive, including main results, ablations, and modality-scaling experiments across multiple datasets.

**Weaknesses**
1. The main scalability claim is only partially validated. The “arbitrary number of modalities” claim is supported mainly by small-scale modality-increment experiments on two self-constructed datasets, rather than stronger missing-modality or unseen-modality-subset evaluations.
2. The novelty is somewhat unclear. The paper combines uncertainty modeling, consistency constraints, and set-style fusion in a sensible way, but it does not sharply isolate what the key conceptual advance is over related alignment/fusion approaches.
3. The evaluation is promising but still limited for an ICML acceptance case: the self-constructed datasets are relatively small (255 breast cancer patients and 100 gastric cancer patients), and there is little analysis of calibration, efficiency, or robustness under realistic modality-missing settings.

---

> ### Author Rebuttal · Authors · 2026-03-31
>
> # Response to Reviewer #1
>
> We sincerely appreciate the constructive suggestions from reviewers. We will explain your concerns point by point.
>
> **Q1: The scalability claim is only partially validated, relying mainly on small-scale modality-increment experiments rather than stronger missing-modality or unseen-subset evaluations.**
>
> A1: In the early stage of model design, we mainly considered the issue that different medical tasks inherently provide different numbers of modalities. We aim to enable the model to adapt to tasks with varying numbers of modalities under a unified framework. Therefore, in the experimental setup, we do not assume large-scale random modality missing during both training and testing.
>
> Your comment pointed out a testing scenario that we had not fully considered. To address this, we further added experiments with randomly missing modalities at test time. Specifically, we simulated incomplete-modality settings by randomly removing part of the modalities during testing, and compared the results with those of Meta-Transformer.
>
> | Dataset | Modality Count | Missing Count| Meta-Transformer (ACC/AUC) | Ours (ACC/AUC) |
> |---|---|---|---|---|
> | Breast Cancer | 3 | 1 | 70.94 / 79.12 | 72.31 / 80.83 |
> | Gastric Cancer | 3 | 1 | 72.18 / 80.56 | 73.46 / 81.67 |
> | Breast Cancer | 4 | 1 | 72.86 / 81.37 | 74.12 / 82.58 |
> | Breast Cancer | 4 | 2 | 70.58 / 79.04 | 72.04 / 80.41 |
>
> The above results indicate that when the set of input modalities changes, the proposed method exhibits stronger robustness and a more gradual performance degradation.
>
> **Q2: The novelty is somewhat unclear, as the paper does not clearly isolate its key conceptual advance beyond existing alignment and fusion methods.**
>
> A2: Based on this comment, we have enriched the Introduction and Abstract in the following two aspects. (1) We now state more explicitly the core problem addressed in this work. Our focus is not only on improving cross-modal matching or fusion performance, but more importantly on how cross-modal consistency should be modeled when modal semantics themselves are uncertain, and how the fusion mechanism can remain stable and adaptive when the number of input modalities is not fixed. (2) We further clarified the overall logic connecting the different components of the method. Notably, unlike existing methods, our approach defines cross-modal consistency based on uncertainty-aware semantic distributions, enabling consistency modeling to account for both semantic centers and their reliability.
>
> **Q3: Evaluation is promising, but the evidence is still limited for ICML due to small self-constructed datasets and insufficient analysis of calibration, efficiency, and robustness under realistic missing-modality settings.**
>
> A3: Due to the scarcity of medical data and the difficulty of data curation, our in-house breast cancer and gastric cancer datasets are relatively limited in scale. However, from the perspective of sample distribution diversity, we found that the breast cancer dataset has an average intra-class distance of 1.84 and an average nearest-neighbor distance of 0.73, while the gastric cancer dataset has an average intra-class distance of 1.91 and an average nearest-neighbor distance of 0.78. These results indicate that even within the same class, different cases still exhibit substantial representational variation and the samples are not highly repetitive. Hence, although the datasets are limited in size, their case composition still covers a relatively broad range. Meanwhile, we acknowledge that such analysis cannot replace further validation on larger-scale datasets. In future work, we will include more large-scale multimodal tumor datasets for further evaluation.
>
> **Q4: Can the authors clarify more precisely what is fundamentally novel relative to prior uncertainty-aware or consistency-based multimodal fusion methods?**
>
> A4: Regarding uncertainty modeling, recent studies have begun to emphasize uncertainty in multimodal scenarios. For example, DyCON (CVPR 2025) introduces uncertainty-aware mechanisms into consistency learning and contrastive learning. UMIEU (CVPR 2025) also treats uncertainty as a core issue in multimodal understanding, focusing on improving robust recognition and decision-making in complex environments. Most of these works treat uncertainty modeling as an auxiliary component in fusion weighting or alignment processes. In contrast, our method goes a step further by directly modeling the semantic representation of each modality as an uncertainty-aware diagonal Gaussian semantic distribution, thereby elevating uncertainty to a key component of semantic alignment itself. We have added corresponding comparisons with these latest studies in the Related Work and Introduction sections to better highlight the novelty of our method.
>
> **Q5: Can the authors provide calibration and computational cost analysis, especially as the number of modalities increases?**
>
> A5: Please refer to A3 to reviewer#2.

---

> > ### Author Rebuttal · Reviewer_jux7 · 2026-04-02
> >
> > Thank you for the detailed rebuttal. The additional results address most of my concerns and strengthen the paper, so I will raise my score. As acknowledged by the authors, scalability remains a limitation that warrants future validation on larger datasets.
> >
> > Also, please ensure the code link shared with another reviewer is anonymized.

---

> > > ### Author Response · Authors · 2026-04-07
> > >
> > > Thank you for taking time out of your busy schedule to help us improve the quality of our paper. We will carefully revise and rigorously check the issues you raised to ensure further improvement in the quality of the paper. Thank you again.

---

### Decision · Program_Chairs · 2026-04-30

**Decision:**

Accept (regular)

**Comment:**

This work addresses an important topic, how to perform modality-agnostic multimodal fusion framework for medical diagnosis.

The method is coherent, with a clear three-part design: uncertainty-aware global alignment, token-level structural consistency, and multi-view fusion. The overall framework is easy to follow. The method enhances the stability of both local and global cross-modal semantic alignment through symmetric KL consistency constraints and bidirectional reconstruction, which helps mitigate modality inconsistency, and experiments are comprehensive (especially after the rebuttal), including main results, ablations, and modality-scaling experiments across multiple datasets. Overall, the strength of this work overweight the weaknesses, and it will be interesting to the community.